# Low-dimensional learned feature spaces quantify individual and group differences in vocal repertoires

**Jack Goffinet[1,2,3], Samuel Brudner[3], Richard Mooney[3], John Pearson[2,3,4,5]***

[1]Department of Computer Science, Duke University, Durham, United States; [2]Center for Cognitive Neurobiology, Duke University, Durham, United States; [3]Department of Neurobiology, Duke University, Durham, United States; [4]Department of Biostatistics & Bioinformatics, Duke University, Durham, United States; [5]Department of Electrical and Computer Engineering, Duke University, Durham, United States

**Abstract** Increases in the scale and complexity of behavioral data pose an increasing challenge for data analysis. A common strategy involves replacing entire behaviors with small numbers of handpicked, domain-specific features, but this approach suffers from several crucial limitations. For example, handpicked features may miss important dimensions of variability, and correlations among them complicate statistical testing. Here, by contrast, we apply the variational autoencoder (VAE), an unsupervised learning method, to learn features directly from data and quantify the vocal behavior of two model species: the laboratory mouse and the zebra finch. The VAE converges on a parsimonious representation that outperforms handpicked features on a variety of common analysis tasks, enables the measurement of moment-by-moment vocal variability on the timescale of tens of milliseconds in the zebra finch, provides strong evidence that mouse ultrasonic vocalizations do not cluster as is commonly believed, and captures the similarity of tutor and pupil birdsong with qualitatively higher fidelity than previous approaches. In all, we demonstrate the utility of modern unsupervised learning approaches to the quantification of complex and high-dimensional vocal behavior.

**\*For correspondence:**
john.pearson@duke.edu

**Competing interest:** The authors declare that no competing interests exist.

## Introduction

Quantifying the behavior of organisms is of central importance to a wide range of fields including ethology, linguistics, and neuroscience. Yet given the variety and complex temporal structure of many behaviors, finding concise yet informative descriptions has remained a challenge. Vocal behavior provides a paradigmatic example: audio data are notoriously high dimensional and complex, and despite intense interest from a number of fields, and significant progress, many aspects of vocal behavior remain poorly understood. A major goal of these various lines of inquiry has been to develop methods for the quantitative analysis of vocal behavior, and these efforts have resulted in several powerful approaches that enable the automatic or semi-automatic analysis of vocalizations (*Tchernichovski and Mitra, 2004*; *Coffey et al., 2019*; *Van Segbroeck et al., 2017*; *Sainburg et al., 2019*; *Tchernichovski et al., 2000*; *Mandelblat-Cerf and Fee, 2014*; *Mets and Brainard, 2018*; *Kollmorgen et al., 2020*; *Holy and Guo, 2005*).

Key to this approach has been the existence of software packages that calculate acoustic features for each unit of vocalization, typically a syllable (*Burkett et al., 2015*; *Tchernichovski and Mitra, 2004*; *Van Segbroeck et al., 2017*; *Coffey et al., 2019*; *Chabout et al., 2015*). For example, Sound Analysis Pro (SAP), focused on birdsong, calculates 14 features for each syllable, including duration, spectral entropy, and goodness of pitch, and uses the set of resulting metrics as a basis for subsequent clustering and analysis (*Tchernichovski and Mitra, 2004*). More recently, MUPET and DeepSqueak have

applied a similar approach to mouse vocalizations, with a focus on syllable clustering (*Van Segbroeck et al., 2017*; *Coffey et al., 2019*). Collectively, these and similar software packages have helped facilitate numerous discoveries, including circadian patterns of song development in juvenile birds (*Derégnaucourt et al., 2005*), cultural evolution among isolate zebra finches (*Fehér et al., 2009*), and differences in ultrasonic vocalizations (USVs) between mouse strains (*Van Segbroeck et al., 2017*).

Despite these insights, this general approach suffers from several limitations: first, handpicked acoustic features are often highly correlated, and these correlations can result in redundant characterizations of vocalization. Second, an experimenter-driven approach may exclude features that are relevant for communicative function or, conversely, may emphasize features that are not salient or capture negligible variation in the data. Third, there is no diagnostic approach to determine when enough acoustic features have been collected: Could there be important variation in the vocalizations that the chosen features simply fail to capture? Lastly and most generally, committing to a syllable-level analysis necessitates a consistent definition of syllable boundaries, which is often difficult in practice. It limits the types of structure one can find in the data and is often difficult to relate to time series such as neural data, for which the relevant timescales are believed to be orders of magnitude faster than syllable rate.

Here, we address these shortcomings by applying a data-driven approach based on variational autoencoders (VAEs) (*Kingma and Welling, 2013*; *Rezende et al., 2014*) to the task of quantifying vocal behavior in two model species: the laboratory mouse (*Mus musculus*) and the zebra finch (*Taeniopygia guttata*). The VAE is an unsupervised modeling approach that learns from data of a pair of probabilistic maps, an 'encoder' and a 'decoder,' capable of compressing the data into a small number of latent variables while attempting to preserve as much information as possible. In doing so, it discovers features that best capture variability in the data, offering a nonlinear generalization of methods like Principal Components Analysis (PCA) and Independent Components Analysis (ICA) that adapts well to high-dimensional data like natural images (*Dai et al., 2018*; *Higgins et al., 2017*). By applying this technique to collections of single syllables, encoded as time-frequency spectrograms, we looked for latent spaces underlying vocal repertoires across individuals, strains, and species, asking whether these data-dependent features might reveal aspects of vocal behavior overlooked by traditional acoustic metrics and provide more principled means for assessing differences among these groups.

Our contributions are fourfold: first, we show that the VAE's learned acoustic features outperform common sets of handpicked features in a variety of tasks, including capturing acoustic similarity, representing a well-studied effect of social context on zebra finch song, and comparing the USVs of different mouse strains. Second, using learned latent features, we report new results concerning both mice and zebra finches, including the finding that mouse USV syllables do not appear to cluster into distinct subtypes, as is commonly assumed, but rather form a broad continuum. Third, we present a novel approach to characterizing stereotyped vocal behavior that does not rely on syllable boundaries, one which we find is capable of quantifying subtle changes in behavioral variability on tens-of-milliseconds timescales. Lastly, we demonstrate that the VAE's learned acoustic features accurately reflect the relationship between songbird tutors and pupils. In all, we show that data-derived acoustic features confirm and extend findings gained by existing approaches to vocal analysis and offer distinct advantages over handpicked acoustic features in several critical applications.

## Results

### VAEs learn a low-dimensional space of vocal features

We trained a VAE *Kingma and Welling, 2013*; *Rezende et al., 2014* to learn a probabilistic mapping between vocalizations and a latent feature space. Specifically, we mapped single-syllable spectrogram images ($D = 16,384$ pixels) to vectors of latent features ($D = 32$) and back to the spectrogram space (*Figure 1a*). As with most VAE methods, we parameterized both the encoder and decoder using convolutional neural networks, which provide useful inductive biases for representing regularly sampled data such as images or spectrograms. The two maps are jointly trained to maximize a lower bound on the probability of the data given the model (see Materials and methods). As in other latent variable models, we assume each observed spectrogram can be explained by an unobserved 'latent' variable situated in some 'latent space.' As visualized in *Figure 1b*, the result is a continuous latent

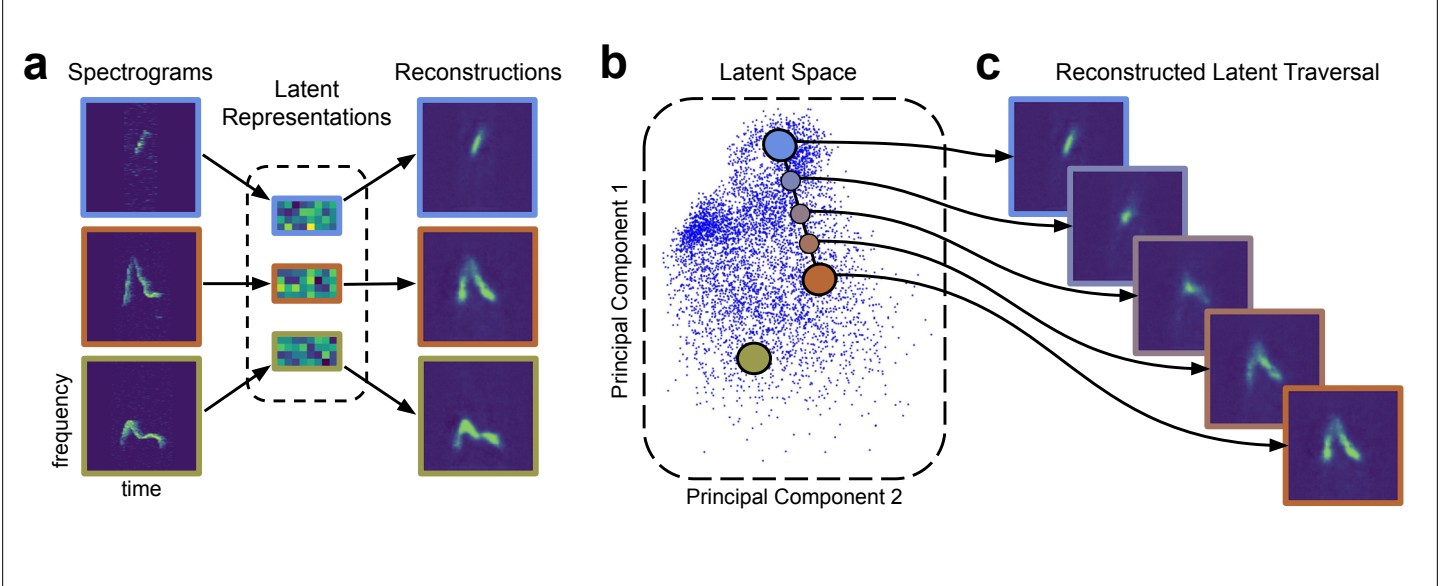

**Figure 1.** Variational autoencoders (VAEs) learn a latent acoustic feature space. (**a**) The VAE takes spectrograms as input (left column), maps them via a probabilistic 'encoder' to a vector of latent dimensions (middle column), and reconstructs a spectrogram via a 'decoder' (right column). The VAE attempts to ensure that these probabilistic maps match the original and reconstructed spectrograms as closely as possible. (**b**) The resulting latent vectors can then be visualized via dimensionality reduction techniques like principal components analysis. (**c**) Interpolations in latent space correspond to smooth syllable changes in spectrogram space. A series of points (dots) along a straight line in the inferred latent space is mapped, via the decoder, to a series of smoothly changing spectrograms (right). This correspondence between inferred features and realistic dimensions of variation is often observed when VAEs are applied to data like natural images (*Kingma and Welling, 2013*; *Rezende et al., 2014*).

The online version of this article includes the following figure supplement(s) for figure 1:

**Figure supplement 1.** Variational autoencoder network architecture.

space that captures the complex geometry of vocalizations. Each point in this latent space represents a single spectrogram image, and trajectories in this latent space represent sequences of spectrograms that smoothly interpolate between start and end syllables (*Figure 1c*). Although we cannot visualize the full 32-dimensional latent space, methods like PCA and the UMAP algorithm *Dai et al., 2018* allow us to communicate results in an informative and unsupervised way. The VAE training procedure can thus be seen as a compression algorithm that represents each spectrogram as a collection of 32 numbers describing data-derived vocal features. In what follows, we will show that these features outperform traditional handpicked features on a wide variety of analysis tasks.

Finally, we note that, while the VAE is compressive—that is, it discards some data—this is both necessary in practice and often desirable. First, necessity: as noted above, nearly all current methods reduce raw audio waveforms to a manageable number of features for purposes of analysis. This is driven in part by the desire to distill these complex sounds into a small collection of interpretable features, but it also stems from the needs of statistical testing, which suffers drastic loss of power for high-dimensional data without advanced methods. Second, this compression, as we will show, is often beneficial as it facilitates visualization and analyses of large collections of vocalizations in new ways. Thus, we view the VAE and its compression-based approach as complementary to both other dimension-reduction techniques like PCA and traditional acoustic signal processing as a method for learning structure from complex data.

## Learned features capture and expand upon typical acoustic features

Most previous approaches to analyzing vocalizations have focused on tabulating a predetermined set of features such as syllable duration or entropy variance that are used for subsequent processing and analysis (*Van Segbroeck et al., 2017*; *Coffey et al., 2019*; *Tchernichovski and Mitra, 2004*; *Burkett et al., 2015*). We thus asked whether the VAE learned feature space simply recapitulated these known features or also captured new types of information missed by traditional acoustic metrics. To address the first question, we trained a VAE on a publicly available collection of mouse USVs (31,440 total

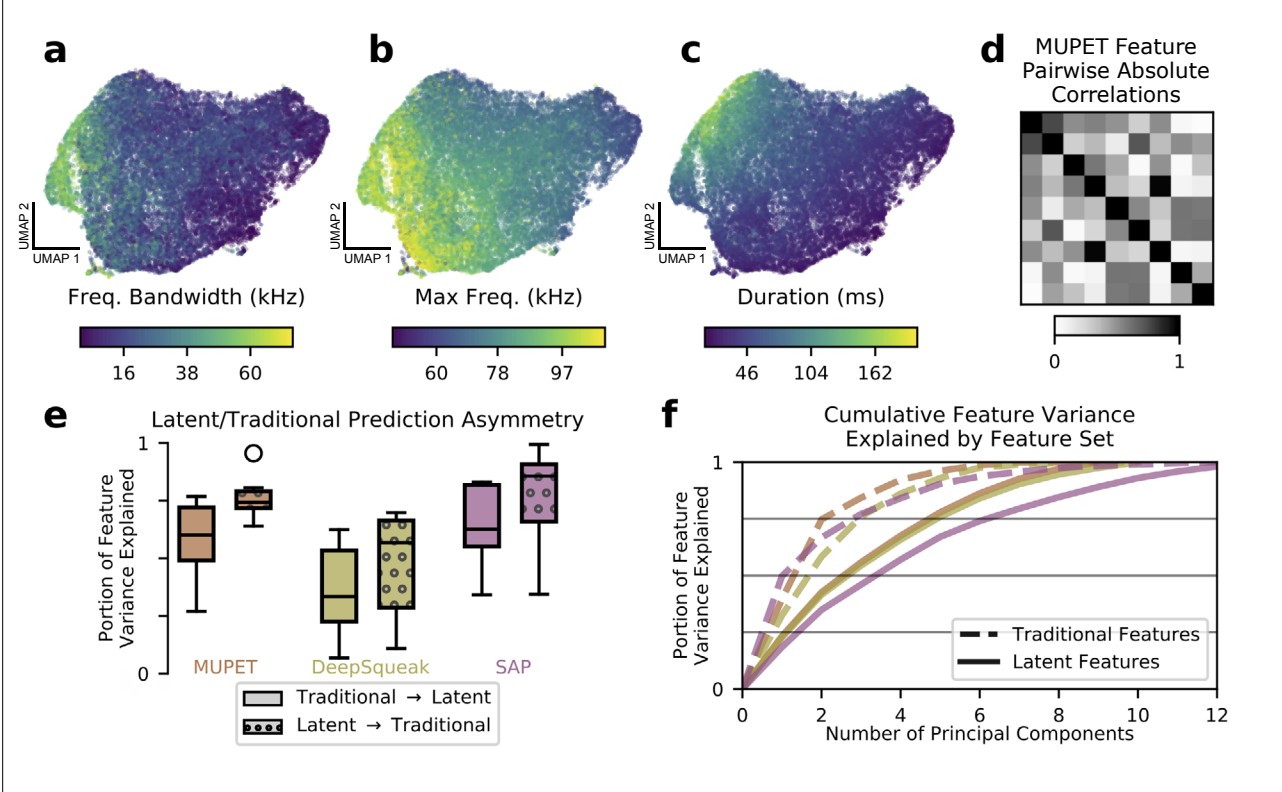

**Figure 2.** Learned acoustic features capture and expand upon traditional features. (**a–c**) UMAP projections of latent descriptions of mouse ultrasonic vocalizations (USVs) colored by three traditional acoustic features. The smoothly varying colors show that these traditional acoustic features are represented by gradients within the latent feature space. (**d**) Many traditional features are highly correlated. When applied to the mouse USVs from (**a**) to (**c**), the acoustic features compiled by the analysis program MUPET have high correlations, effectively reducing the number of independent measurements made. (**e**) To better understand the representational capacity of traditional and latent acoustic features, we used each set of features to predict the other and vice versa (see Materials and methods). We find that, across software programs, the learned latent features were better able to predict the values of traditional features than vice versa, suggesting that they have a higher representational capacity. Central line indicates median, upper and lower box the 25th and 75th percentiles, respectively. Whiskers indicate 1.5 times the interquartile range. Feature vector dimensions: MUPET, 9; DeepSqueak, 10; SAP, 13; mouse latent, 7; zebra finch latent, 5. (**f**) As another test of representational capacity, we performed PCA on the feature vectors to determine the effective dimensionality of the space spanned by each set of features (see Materials and methods). We find in all cases that latent features require more principal components to account for the same portion of feature variance, evidence that latent features span a higher dimensional space than traditional features applied to the same datasets. Colors are as in (**e**). Latent features with colors labeled 'MUPET' and 'DeepSqueak' refer to the *same* set of latent features, truncated at different dimensions corresponding to the number of acoustic features measured by MUPET and DeepSqueak, respectively.

The online version of this article includes the following figure supplement(s) for figure 2:

**Figure supplement 1.** Variance explained by traditional and latent features.

**Figure supplement 2.** The variational autoencoder (VAE) learns a parsimonious set of acoustic features.

**Figure supplement 3.** Correlations among traditional and latent features.

**Figure supplement 4.** Reproducibility of variational autoencoder (VAE) latent features.

**Figure supplement 5.** The effect of time stretch and frequency spacing parameters.

**Figure supplement 6.** Removing noise from single mouse ultrasonic vocalization (USV) recordings (see Recordings).

syllables [*Van Segbroeck M et al., 2019*]), inferred latent features for each syllable, and colored the results according to three acoustic features—frequency bandwidth, maximum frequency, and duration—calculated by the analysis program MUPET (*Van Segbroeck et al., 2017*). As *Figure 2a–c* shows, each acoustic feature appears to be encoded in a smooth gradient across the learned latent space, indicating that information about each has been preserved. In fact, when we quantified this pattern by asking how much variance in a wide variety of commonly used acoustic metrics could be accounted for by latent features (see Materials and methods), we found that values ranged from 64%

to 95%, indicating that most or nearly all traditional features were captured by the latent space (see *Figure 2—figure supplement 1* for individual acoustic features). Furthermore, we found that, when the analysis was reversed, commonly used acoustic features were not able to explain as much variance in the VAE latent features, indicating a prediction asymmetry between the two sets (*Figure 2e*). That is, the learned features carry most of the information available in traditional features, as well as unique information missed by those metrics.

We thus attempted to compare the effective representational capacity of the VAE to current best approaches in terms of the dimensionalities of their respective feature spaces. We begin by noting that the VAE, although trained with a latent space of 32 dimensions, converges on a parsimonious representation that makes use of only 5–7 dimensions, with variance apportioned roughly equally between these (*Figure 2—figure supplement 2*; *Dai et al., 2018*). For the handpicked features, we normalized each feature independently by z-score to account for scale differences. For comparison purposes, we applied the same normalization step to the learned features, truncated the latent dimension to the number of handpicked features, and calculated the cumulative feature variance as a function of number of principal components (*Figure 2f*). In such a plot, shallow linear curves are preferred since this indicates that variance is apportioned roughly equally among principal components and the effective dimensionality of the space is large. Equivalently, this means that the eigenvalue spectrum of the feature correlation matrix is close to the identity. As *Figure 2f* thus makes clear, the spaces spanned by the learned latent features have comparatively higher effective dimension than the spaces spanned by traditional features, suggesting that the learned features have a higher representational capacity. While the three software packages we tested (SAP [*Tchernichovski and Mitra, 2004*], MUPET [*Van Segbroeck et al., 2017*], DeepSqueak [*Coffey et al., 2019*]) measure upwards of 14 acoustic features per syllable, we find that these features often exhibit high correlations (*Figure 2d*, *Figure 2—figure supplement 3*), effectively reducing the number of independent measurements made. While correlations among features are not necessarily undesirable, they can complicate subsequent statistical testing because nonlinear relationships among features violate the assumptions of many statistical tests. VAE features, which allow for nonlinear warping, avoid this potential difficulty.

The degree to which the learned features capture novel information can also be demonstrated by considering their ability to encode a notion of spectrogram similarity since this is a typical use to which they are put in clustering algorithms (although see *Van Segbroeck et al., 2017* for an alternative approach to clustering). We tested this by selecting query spectrograms and asking for the closest spectrograms as represented in both the DeepSqueak acoustic feature space and the VAE's learned latent space. As *Figure 3* shows, DeepSqueak feature space often fails to return similar spectrograms, whereas the learned latent space reliably produces close matches (see *Figure 3—figure supplement 1* for comparisons to metrics in spectrogram space, *Figure 3—figure supplement 2* for a representative sample using all feature sets, and *Figure 3—figure supplement 3* for more details on nearest

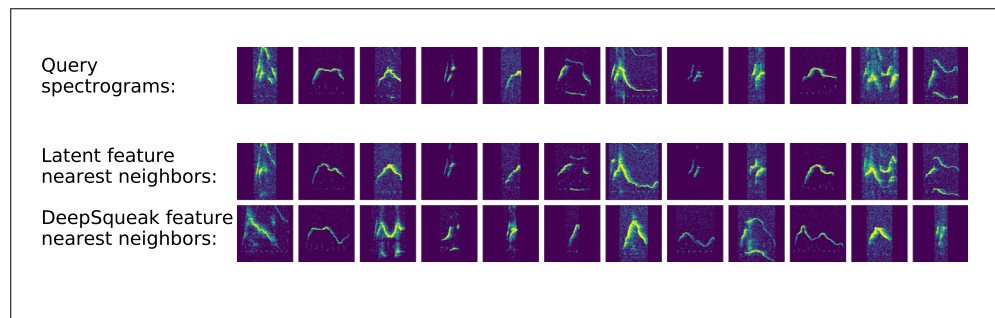

**Figure 3.** Latent features better represent acoustic similarity. Top row: example spectrograms; middle row: nearest neighbors in latent space; bottom row: nearest neighbors in DeepSqueak feature space.

The online version of this article includes the following figure supplement(s) for figure 3:

**Figure supplement 1.** Nearest neighbors returned by distance metrics in spectrogram space exhibit failure modes not found in latent space.

**Figure supplement 2.** Representative sample of nearest neighbors returned by several feature spaces.

**Figure supplement 3.** Investigating poor DeepSqueak feature nearest neighbors from *Figure 3*.

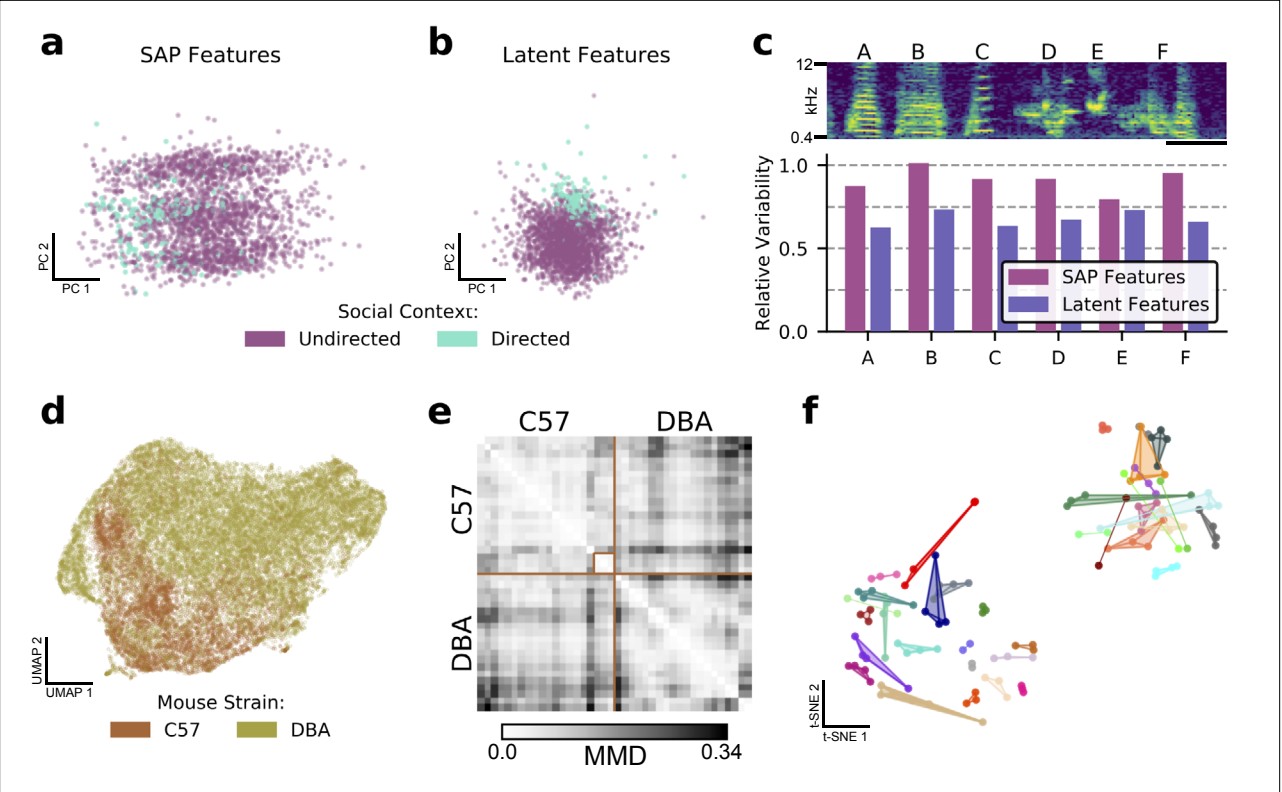

**Figure 4.** Latent features better capture differences in sets of vocalizations. (**a**) The first two principal components in SAP feature space of a single zebra finch song syllable, showing differences in directed and undirected syllable distributions. (**b**) The first two principal components of latent syllable features, showing the same comparison. Learned latent features more clearly indicate differences between the two conditions by clustering directed syllables together. (**c**) Acoustic variability of each song syllable as measured by SAP features and latent features (see Methods). Latent features more clearly represent the constriction of variability in the directed context. Spectrogram scale bars denote 100ms. (**d**) A UMAP projection of the latent means of USV syllables from two strains of mice, showing clear differences in their vocal repertoires. (**e**) Similarity matrix between syllable repertoires for each of the 40 reccording sessions from (**d**). Lighter values correspond to more similar syllable repertoires (lower Maximum Mean Discrepancy (MMD)). (**f**) t-SNE representation of similarities between syllable repertoires, where distance metric is estimated MMD. The dataset, which is distinct from that represented in (d) and (e), contains 36 individuals, 118 recording sessions, and 156,180 total syllables. Color indicates individual mice and scatterpoints of the same color represent repertoires recorded on different days. Distances between points represent the similarity in vocal repertoires, with closer points more similar. We note that the major source of repertoire variability corresponds to genetic background, corresponding to the two distinct clusters (*Figure 4—figure supplement 3*). A smaller level of variability can be seen across individuals in the same clusters. Individual mice have repertoires with even less variability, indicated by the close proximity of most repertoires for each mouse.

The online version of this article includes the following figure supplement(s) for figure 4:

**Figure supplement 1.** Latent features better represent constricted variability of female-directed zebra finch song.

**Figure supplement 2.** An 'atlas' of mouse ultrasonic vocalizations (USVs).

**Figure supplement 3.** Details of *Figure 4f*.

---

neighbors returned by DeepSqueak feature space). This suggests that the learned features better characterize local variation in the data by more accurately arranging nearest neighbors.

## Latent spaces facilitate comparisons between vocal repertoires

Many experimental designs require quantifying differences between sets of vocalizations. As a result, the ability of a feature set to distinguish between syllables, individuals, and groups poses a key test of the VAE-based approach. Here, we apply the VAE latent features to several comparison problems for which handpicked features are often used.

A common comparison in birdsong research is that between female-directed and undirected song. It is well-established that directed song is more stereotyped and slightly faster than undirected song (*Sossinka and Böhner, 1980*). We thus asked whether the learned features could detect this effect. In *Figure 4a*, we plot the first two principal components of acoustic features calculated by the Sound

Analysis Pro software package (*Tchernichovski and Mitra, 2004*) for both directed and undirected renditions of a single zebra finch song syllable. We note a generally diffuse arrangement and a subtle leftward bias in the directed syllables compared to the undirected syllables. *Figure 4b* displays the same syllables with respect to the first two principal components of the VAE's latent features, showing a much more concentrated distribution of directed syllables relative to undirected syllables (see *Figure 4—figure supplement 1* for all syllables). In fact, when we quantify this reduction of variability across all feature-space dimensions and song syllables (see Materials and methods), learned latent features consistently report greater variability reductions than SAP-generated features (*Figure 4c*; SAP: 0–20%, VAE: 27–37%), indicating that latent features are more sensitive to this effect. Additionally, we find that latent features outperform SAP features in the downstream tasks of predicting social context (*Appendix 1—table 1*), mouse strain (*Appendix 1—table 2*), and mouse identity (*Appendix 1—table 3*).

Similarly, we can ask whether latent features are able to capture differences between groups of individuals. In *Van Segbroeck et al., 2017*, the authors compared USVs of 12 strains of mice using a clustering-based approach. Here, we perform an alternative version of this analysis using two mouse strains (C57/BL6 and DBA/2) from a publicly available dataset that were included in this earlier study. *Figure 4d* shows a UMAP projection of the 31,440 detected syllables, colored by mouse strain. Visualized with UMAP, clear differences between the USV distributions are apparent. While in contrast to traditional acoustic features such as 'mean frequency' individual VAE latent features (vector components) are generally less interpretable, when taken together with an 'atlas' of USV shapes derived from this visualization (*Figure 4—figure supplement 2*), we can develop an intuitive understanding of the differences between the USVs of the two strains: the C57 mice mostly produce noisy USVs, while the DBA mice produce a much greater variety, including many short low-frequency syllables that C57s rarely produce.

Given these results, we asked whether these strain differences are evident at the level of individual 6.5 min recording sessions. To compare distributions of syllables without making restrictive parametric assumptions, we employed maximum mean discrepancy (MMD), a difference measure between pairs of distributions (*Gretton et al., 2012*). We estimated MMD between the distributions of latent syllable encodings for each pair of recording sessions (see Materials and methods) and visualized the result as a distance matrix (*Figure 4e*). Here, lighter values indicate more similar syllable repertoires. We note that, in general, values are brighter when comparing repertoires within strains than when comparing across strains, consistent with the hypothesis of inter-strain differences. We also note some substructure, including a well-defined cluster within the C57 block (annotated).

Finally, we used a much larger library of female-directed mouse USVs (36 individuals, 2–4 20-min recording sessions each, 40 total hours of audio, 156,000 syllables) to investigate the diversity and stability of syllable repertoires. We repeated the above procedure, estimating MMD for each pair of recording sessions (*Figure 4—figure supplement 3*), and then computed a t-distributed Stochastic Neighbor Embedding t-SNE layout of the recording sessions with estimated MMD as the distance metric to visualize the distribution of syllable repertoires (see Materials and methods). In *Figure 4f*, each recording session is represented by a scatterpoint, and recordings of the same individual are connected and displayed in the same color. We note an overall organization of syllables into two clusters, corresponding to the genetic backgrounds of the mice (*Figure 4—figure supplement 3*). Furthermore, we note that almost all recordings of the same individuals are co-localized, indicating that within-subject differences in syllable repertoire are smaller than those between individuals. Although it has been previously shown that a deep convolutional neural network can be trained to classify USV syllables according to mouse identity with good accuracy (*Ivanenko et al., 2020*, Figure S1), here we find that repertoire features learned in a wholly unsupervised fashion achieve similar results, indicating that mice produce individually stereotyped, stable vocal repertoires.

## Latent features fail to support cluster substructure in USVs

Above, we have shown that, by mapping complex sets of vocalizations to low-dimensional latent representations, VAEs allow us to visualize the relationships among elements in mouse vocal repertoires. The same is likewise true for songbirds such as the zebra finch, *T. guttata*. *Figure 5* compares the geometry of learned latent spaces for an individual of each species as visualized via UMAP. As expected, the finch latent space exhibits well-delineated clusters corresponding to song syllables

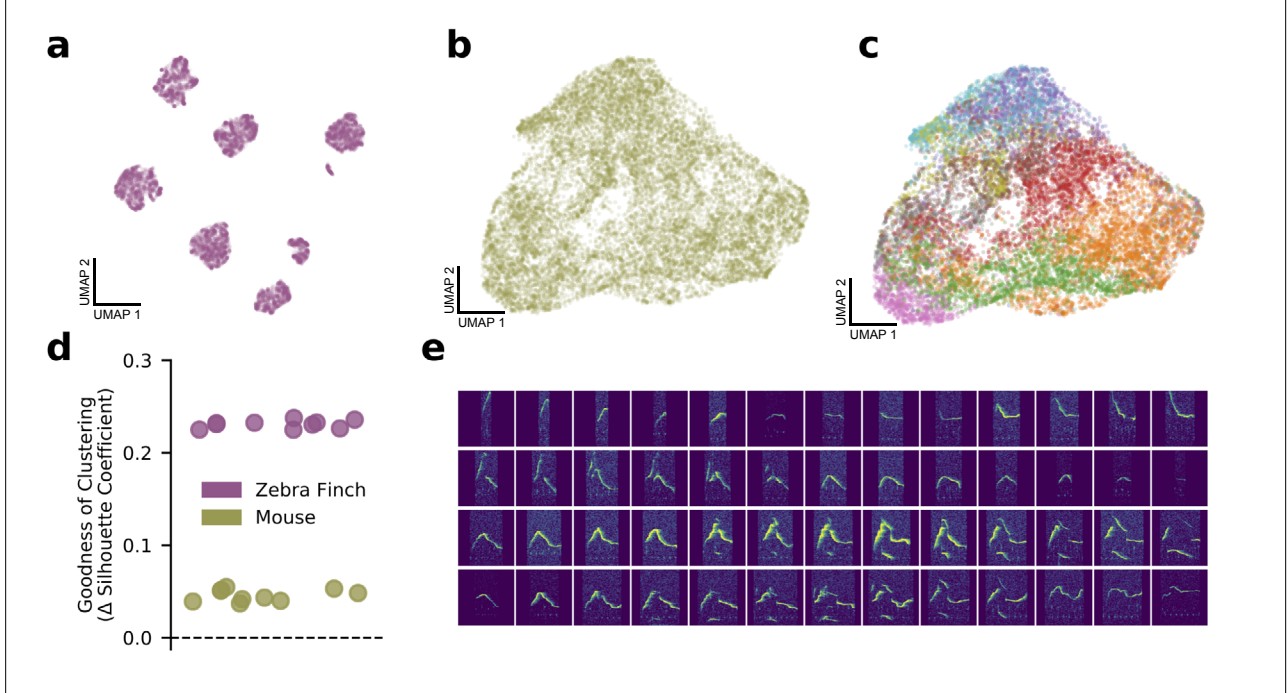

**Figure 5.** Bird syllables clearly cluster, but mouse ultrasonic vocalizations (USVs) do not. (**a**) UMAP projection of the song syllables of a single male zebra finch (14,270 syllables). (**b**) UMAP projection of the USV syllables of a single male mouse (17,400 syllables). (**c**) The same UMAP projection as in (**b**), colored by MUPET-assigned labels. (**d**) Mean silhouette coefficient (an unsupervised clustering metric) for latent descriptions of zebra finch song syllables and mouse syllables. The dotted line indicates the null hypothesis of a single covariance-matched Gaussian noise cluster fit by the same algorithm. Each scatterpoint indicates a cross-validation fold, and scores are plotted as differences from the null model. Higher scores indicate more clustering. (**e**) Interpolations (horizontal series) between distinct USV shapes (left and right edges) demonstrating the lack of data gaps between putative USV clusters.

The online version of this article includes the following figure supplement(s) for figure 5:

**Figure supplement 1.** Evaluation of clustering metrics on vocalization for different cluster numbers.

**Figure supplement 2.** Evaluation of clustering metrics on different mouse ultrasonic vocalization (USV) feature sets.

**Figure supplement 3.** Evaluation of clustering metrics on different zebra finch syllable feature sets.

**Figure supplement 4.** Reliability of clustering for zebra finch syllables and mouse ultrasonic vocalizations (USVs).

**Figure supplement 5.** Absence of continuous interpolations between zebra finch song syllables.

(*Figure 5a*). However, as seen above, mouse USVs clump together in a single quasi-continuous mass (*Figure 5b*). This raises a puzzle since the clustering of mouse vocalizations is often considered well-established in the literature (*Holy and Guo, 2005*; *Burkett et al., 2015*; *Woehr, 2014*; *Chabout et al., 2015*; *Hertz et al., 2020*) and is assumed in most other analyses of these data (*Van Segbroeck et al., 2017*; *Coffey et al., 2019*). Clusters of mouse USVs are used to assess differences across strains (*Van Segbroeck et al., 2017*), social contexts (*Chabout et al., 2015*; *Coffey et al., 2019*; *Hammer-schmidt et al., 2012*), and genotypes (*Gaub et al., 2010*), and the study of transition models among clusters of syllables has given rise to models of syllable sequences that do not readily extend to the nonclustered case (*Holy and Guo, 2005*; *Chabout et al., 2015*; *Hertz et al., 2020*).

We therefore asked whether mouse USVs do, in fact, cluster or whether, as the latent space projection suggests, they form a single continuum. In principle, this is impossible to answer definitively because, without the benefit of ground truth labels, clustering is an unsupervised classification task. Moreover, there is little consensus among researchers as to the best method for assessing clustering and where the cutoff between clustered and nonclustered data lies (*Jain et al., 1999*). In practice, new clustering algorithms are held to function well when they outperform previous approaches and produce sensible results on data widely agreed on to be clustered. Thus, while it is clear that zebra finch song syllables should be and are clustered by the VAE (*Figure 5a*), we can only ask whether clustering is a more or less satisfying account of the mouse data in *Figure 5b*.

To address this question, we performed a series of analyses to examine the clustering hypothesis from complementary angles. First, we asked how clusters detected by other analysis approaches correspond to regions in the latent space. As shown in *Figure 5c*, clusters detected by MUPET roughly correspond to regions of the UMAP projection, with some overlap between clusters (e.g., purple and blue clusters) and some noncontiguity of single clusters (red and orange clusters). That is, even though clusters do broadly label different subsets of syllables, they also appear to substantially bleed into one another, unlike the finch song syllables in *Figure 5a*. However, it might be objected that *Figure 5b* displays the UMAP projection, which only attempts to preserve local relationships between nearest neighbors and is not to be read as an accurate representation of the latent geometry. Might the lack of apparent clusters result from distortions produced by the projection to two dimensions? To test this, we calculated several unsupervised clustering metrics on full, unprojected latent descriptions of zebra finch and mouse syllables. By these measures, both bird syllables and mouse USVs were more clustered than moment-matched samples of Gaussian noise, a simple null hypothesis, but mouse USVs were closer to the null than to birdsong on multiple goodness-of-clustering metrics (*Figure 5d*, *Figure 5—figure supplement 1*). Additionally, we find that MUPET acoustic features admit uniformly poorer clusters than latent features as quantified by the same metrics (*Figure 5—figure supplements 2 and 3*). On a more practical note, we compared the consistency of cluster labels assigned by Gaussian mixture models (GMMs) trained on disjoint subsets of zebra finch and mouse latent syllable descriptions to determine how well the structure of the data determines cluster membership across repeated fittings (*Figure 5—figure supplement 4*). We find near-perfectly consistent assignments of zebra finch syllables into six clusters and much less consistent clusters for mouse syllables for more than two clusters. Finally, we tested whether the data contained noticeable gaps between syllables in different clusters. If syllable clusters are well-defined, there should not exist smooth sequences of datapoints connecting distinct examples. However, we find that even the most acoustically disparate syllables can be connected with a sequence of syllables exhibiting more-or-less smooth acoustic variation (*Figure 5e*), in contrast to zebra finch syllables (*Figure 5—figure supplement 5*). Thus, even though clustering may not constitute the best account of mouse USV syllable structure, learned latent features provide useful tools to both explore and quantify the acoustic variation within and across species.

## Measuring acoustic variability over tens of milliseconds

The results above have shown that data-derived latent features represent more information about syllables than traditional metrics and can successfully capture differences within and between individuals and groups. Here, we consider how a related approach can also shed light on the short-time substructure of vocal behavior.

The analysis of syllables and other discrete segments of time is limited in at least two ways. First, timing information, such as the lengths of gaps between syllables, is ignored. Second, experimenters must choose the unit of analysis (syllable, song motif, bout), which has a significant impact on the sorts of structure that can be identified (*Kershenbaum et al., 2016*). In an attempt to avoid these limitations, we pursued a complementary approach, using the VAE to infer latent descriptions of fixed duration audio segments, irrespective of syllable boundaries. Similar to the shotgun approach to gene sequencing (*Venter et al., 1998*) and a related method of neural connectivity inference (*Soudry et al., 2015*), we trained the VAE on randomly sampled segments of audio, requiring that it learn latent descriptions sufficient to characterize any given time window during the recording. That is, this 'shotgun-VAE' approach encouraged the autoencoder to find latent features sufficient to 'glue' continuous sequences back together from randomly sampled audio snippets.

*Figure 6a* shows a UMAP projection of latent features inferred from fixed-duration segments from a subset of the mouse USVs shown in *Figure 5b*. While this projection does display some structure (silence on the right, shorter to longer syllables arranged from right to left), there is no evidence of stereotyped sequential structure (see also *Figure 6—videos 1* and *2*). In contrast, *Figure 6b* shows the same technique applied to bouts of zebra finch song, with the song represented as a single well-defined strand coursing clockwise from the bottom to the top left of the projection. Other notable features are the loop on the left containing repeated, highly variable introductory notes that precede and often join song renditions and a 'linking note' that sometimes joins song motifs. Most importantly,

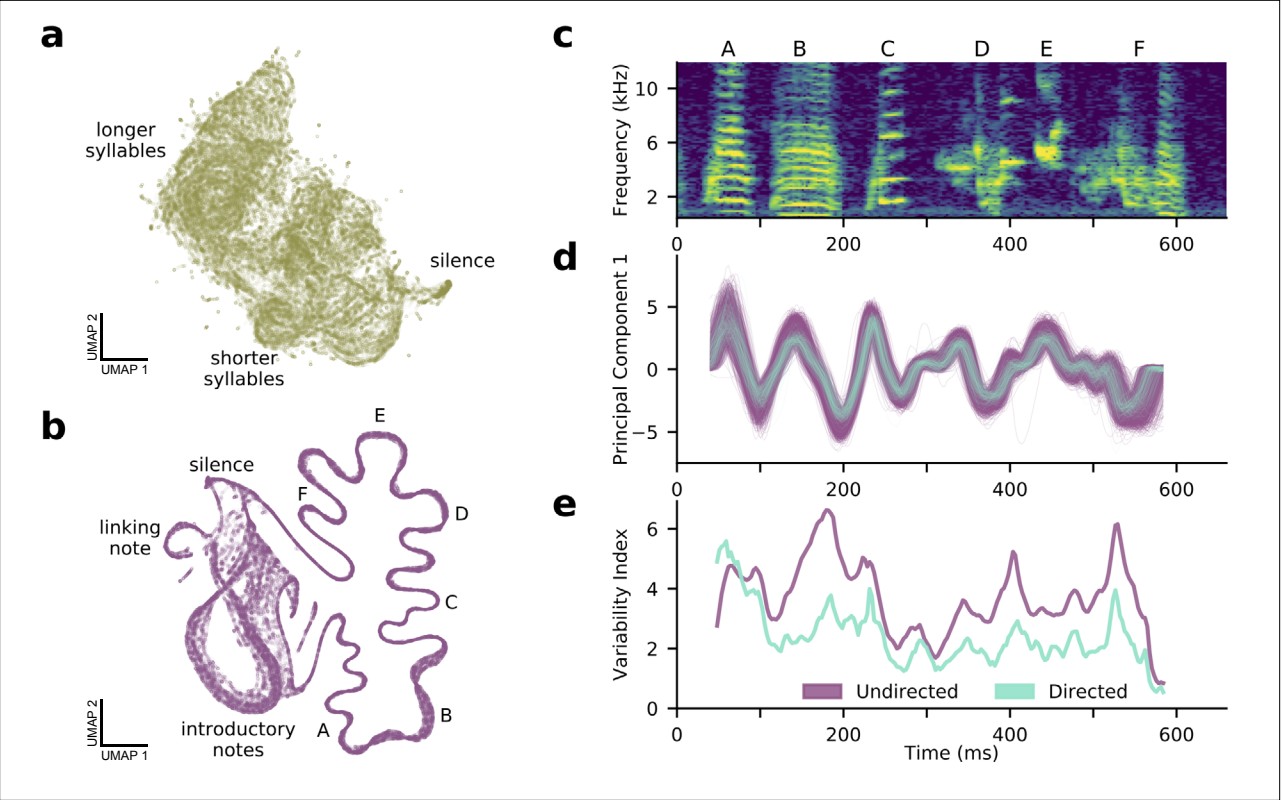

**Figure 6.** A shotgun variational autoencoder approach learns low dimensional latent representations of subsampled, fixed-duration spectrograms and captures short-timescale variability in behavior. (**a**) A UMAP projection of 100,000 200-ms windows of mouse ultrasonic vocalizations (cp. *Figure 4a*). (**b**) A UMAP projection of 100,000 120-ms windows of zebra finch song (cp. *Figure 4b*). Song progresses counterclockwise on the right side, while more variable, repeated introductory notes form a loop on the left side. (**c**) A single rendition of the song in (**b**). (**d**) The song's first principal component in latent space, showing both directed (cyan) and undirected (purple) renditions. (**e**) In contrast to a syllable-level analysis, the shotgun approach can measure zebra finch song variability in continuous time. Song variability in both directed (cyan) and undirected (purple) contexts is plotted (see Materials and methods).

The online version of this article includes the following video and figure supplement(s) for figure 6:

**Figure 6—video 1.** An animated version of Figure 6b.

https://elifesciences.org/articles/67855/figures#fig6video1

**Figure 6—video 2.** An animated version of Figure 6a.

https://elifesciences.org/articles/67855/figures#fig6video2

**Figure supplement 1.** Effect of window duration on shotgun variational autoencoder (VAE).

**Figure supplement 2.** Effect of time warping on shotgun variational autoencoder.

such a view of the data clearly illustrates not only stereotypy but variability: introductory notes are highly variable, but so are particular syllables (**B**, **E**) in contrast to others (**C**, **F**).

Following this, we asked whether the shotgun VAE method could be used to assess the phenomenon of reduced variability in directed birdsong (*Sossinka and Böhner, 1980*). We examined the song portion of *Figure 6b* in both directed and undirected conditions, warping each in time to account for well-documented differences in rhythm and tempo. We then trained a VAE on randomly sampled 80 ms portions of the warped spectrograms. As a plot of the first principal component of the latent space shows (*Figure 4d*), the VAE is able to recover the expected reduction in directed song variability on a tens-of-milliseconds timescale relevant to the hypothesized neural underpinnings of the effect (*Fee and Goldberg, 2011*). This result recapitulates similar analyses that have focused on harmonic and tonal syllables like A and B in *Figure 4c* (*Kao and Brainard, 2006*), but the shotgun VAE method is applicable to all syllables, yielding a continuous estimate of song variability (*Figure 4e*). Thus, not only do VAE-derived latent features capture structural properties of syllable repertoires, the shotgun VAE approach serves to characterize continuous vocal dynamics as well.

## Latent features capture song similarity

Above, we saw how a subsampling-based 'shotgun VAE' approach can capture fine details of zebra finch vocal behavior modulated by social context. However, a principal reason songbirds are studied is their astonishing ability to copy song. A young male zebra finch can successfully learn to sing the song of an older male over the course of multiple months after hearing only a handful of song renditions. At least three methods exist for quantifying the quality of song learning outcomes, with two using handpicked acoustic features (*Tchernichovski et al., 2000*; *Tchernichovski and Mitra, 2004*; *Mandelblat-Cerf and Fee, 2014*) and another using Gaussian distributions in a similarity space based on power spectral densities to represent syllable categories (*Mets and Brainard, 2018*). We reiterate that handpicked acoustic features are sensitive to experimenter choices, with some acoustic features like pitch only defined for certain kinds of sounds. Additionally, restrictive parametric assumptions limit the potential sensitivity of a method. By contrast, the VAE learns a compact feature representation of complete spectrograms and MMD provides a convenient nonparametric difference measure between distributions. Therefore, as a final assessment of the VAE's learned features, we asked whether latent features reflect the similarity of tutor and pupil songs.

To assess the VAE's ability to capture effects of song learning, we trained a syllable VAE and a shotgun VAE on song motifs of 10 paired adult zebra finch tutors and adult zebra finch pupils. As *Figure 7a* shows for the syllable-level analysis, most syllables form distinct clusters in a latent UMAP embedding, with many tutor/pupil syllable pairs sharing a cluster (*Figure 7—video 1*). For a specific tutor/pupil pair shown in *Figure 7b*, we highlight three corresponding syllables from the two motifs. The first and third syllables (C and E) are well-copied from tutor to pupil, but the pupil's copy of the second syllable (syllable D) does not contain the high-frequency power present in the first half of the tutor's syllable. This discrepancy is reflected in the latent embedding, with the two versions of syllable D corresponding to distinct clusters. We quantified the quality of copying by calculating MMD between each pair of tutor and pupil syllables, shown in *Figure 7c*. A dark band of low MMD values along the diagonal indicates well-copied syllables and syllable orderings for most birds.

To complement the previous analysis, we performed an analogous analysis using the shotgun VAE approach. After training the VAE on 60 ms chunks of audio drawn randomly from each song motif, we projected the learned latent features into two dimensions using UMAP (*Figure 7d*) with a modified distance to prevent motif 'strands' from breaking (see Materials and methods, *Figure 7—figure supplement 1*). As expected, we see a close correspondence between pupil and tutor motif strands, with only a single tutor/pupil pair (pink) not overlapping. In fact, out of the roughly 3.6 million possible pairings of pupils and tutors, it is easily apparent which is the true pairing. Additionally, we find that the finer details are informative. For instance, the poorly copied syllable 'D' from *Figure 7b* corresponds to a temporary divergence of the pupil strand from the tutor strand, reflected by large MMD values for both shotgun and syllable analyses (*Figure 7e*). Additionally, we find that the shotgun VAE approach accurately judges the pupil's fused 'AB' syllable to be similar to the tutor's 'A' and 'B' syllables, in contrast to the syllable-level analysis (*Figure 7e*, bottom). We quantified the quality of copying in continuous time for all tutor/pupil pairs by calculating MMD between the distributions of song latents corresponding to distinct times within motifs. Deviations from a simple rank-one structure are shown in *Figure 7f* (see *Figure 7—figure supplement 2* for the original MMD matrix). Consistent with *Figure 7c*, a dark band near the diagonal indicates good copying, quantifying the quality of copying in much more detail than a syllable-level analysis could provide.

## Discussion

The complexity and high dimensionality of vocal behavior have posed a persistent challenge to the scientific study of animal vocalization. In particular, comparisons of vocalizations across time, individuals, groups, and experimental conditions require some means of characterizing the similarity of selected groups of vocal behaviors. Feature vector-based approaches and widespread software tools have gone a long way toward addressing this challenge and providing meaningful scientific insights, but the reliance of these methods on handpicked features leaves open the question of whether other feature sets might better characterize vocal behavior.

Here, we adopt a data-driven approach, demonstrating that features learned by the VAE, an unsupervised learning method, outperform frequently used acoustic features across a variety of common

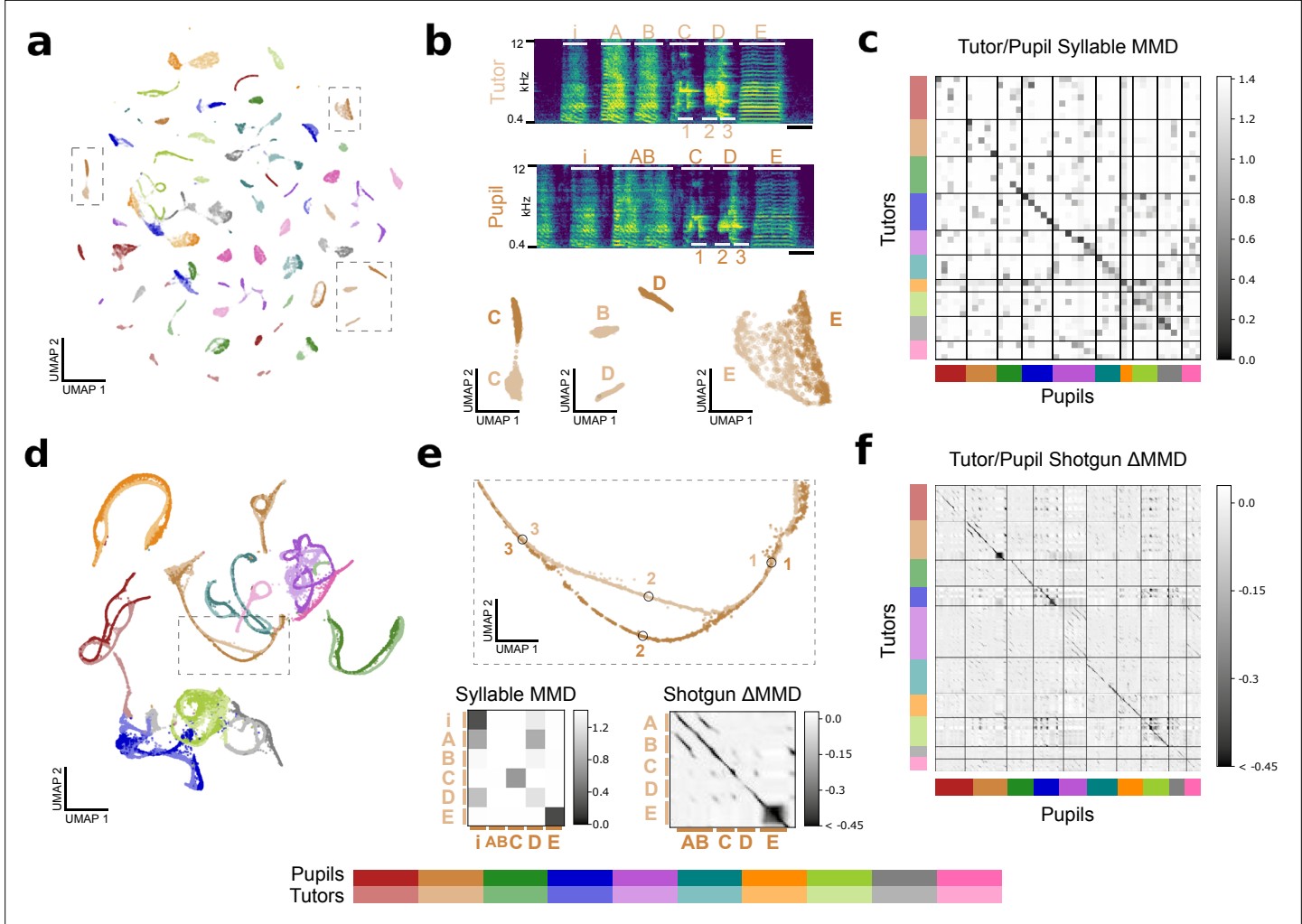

**Figure 7.** Latent features capture similarities between tutor and pupil song. (**a**) Latent UMAP projection of the song syllables of 10 zebra finch tutor/pupil pairs. Note that many tutor and pupil syllables cluster together, indicating song copying. (**b**) Example song motifs from one tutor/pupil pair. Letters A-E indicate syllables. Syllables Syllables "C" and "E" are well-copied, but the pupil's rendition of syllable "D" does not have as much high-frequency power as the tutor's rendition. The difference between these two renditions is captured in the latent UMAP projection below. Additionally, the pupil sings a concatenated version of the tutor's syllables "A" and "B," a regularity that a syllable-level analysis cannot capture. Thus, the pupil's syllable "AB" does not appear near tutor syllable "B" in the UMAP projection. Scale bar denotes 100ms. (**c**) Quality of song copying, estimated by maximum mean discrepancy (MMD) between every pair of tutor and pupil syllables. The dark band of low MMD values near the diagonal indicates good song copying. Syllables are shown in motif order. (**d**) Latent UMAP projection of shotgun VAE latents (60ms windows) for the song motifs of 10 zebra finch tutor/pupil pairs. Song copying is captured by the extent to which pupil and tutor strands co-localize. (**e**) Top: Detail of the UMAP projection in panel d shows a temporary split in pupil and tutor song strands spanning the beginning of syllable "D" with poorly copied high-frequency power. Labeled points correspond to the motif fragments marked in panel b. Bottom: Details of the syllable and shotgun MMD matrices in panels c and f. Note high MMD values for the poorly copied "D" syllable. Additionally, the syllable-level analysis reports high MMD betwen the pupil's fused "AB" and syllable and tutor's"A" and "B" syllables, though the shotgun VAE approach reports high similarity (low MMD) between pupil andtutor throughout these syllables. (**f**) MMD between pupil and tutor shotgun VAE latents indexed by continuous-valued time-in-motif quantifies song copying on fine timescales. The dark bands near the diagonal indicate well-copied stretches of song. The deviation of MMD values from a rank-one matrix is displayed for visual clarity (see *Figure 7—figure supplement 2* for details). Best viewed zoomed in.

The online version of this article includes the following video and figure supplement(s) for figure 7:

**Figure supplement 1.** Effect of modified UMAP distance matrix on shotgun variational autoencoder (VAE) from *Figure 7d,e*.

**Figure supplement 2.** Full and rank-one estimates of maximum mean discrepancy (MMD) matrix.

**Figure 7—video 1.** An animated version of Figure 7d.

https://elifesciences.org/articles/67855/figures#fig7video1

analysis tasks. As we have shown, these learned features are both more parsimonious (*Figure 2— figure supplement 3*), capture more variability in the data (*Figure 2e and f*), and better characterize vocalizations as judged by nearest neighbor similarity (*Figure 3*, *Figure 3—figure supplement 1*, *Figure 3—figure supplement 2*). Moreover, these features easily facilitate comparisons across sessions (*Figure 4f*), social contexts (*Figure 4a–c*), and individuals (*Figures 4d–f and 7*), quantifying not only differences in mean vocal behavior (*Figure 4d*), but also in vocal variability (*Figure 4c*).

This data-driven approach is closely related to previous studies that have applied dimensionality reduction algorithms (UMAP [*McInnes et al., 2018*] and t-SNE [*Lvd and Hinton, 2008*]) to spectrograms to aid in syllable clustering of birdsong (*Sainburg et al., 2019*) and visualize juvenile song learning in the zebra finch (*Kollmorgen et al., 2020*). Additionally, a related recent publication (*Sainburg et al., 2020*) similarly described the application of UMAP to vocalizations of several more species and the application of the VAE to generate interpolations between birdsong syllables for use in playback experiments. Here, by contrast, we restrict use of the UMAP and t-SNE dimensionality reduction algorithms to visualizing latent spaces inferred by the VAE and use the VAE as a general-purpose tool for quantifying vocal behavior, with a focus on cross-species comparisons and assessing variability across groups, individuals, and experimental conditions.

Moreover, we have argued above that, despite conventional wisdom, clustering is not the best account of the diversity of mouse vocal behavior. We argued this on the basis of multiple converging lines of evidence, but note three important qualifications: first, the huge variety of vocal behavior among rodents (*Berryman, 1976*; *Holy and Guo, 2005*; *Novakowski, 1969*; *Sadananda et al., 2008*; *Smith et al., 1977*; *Miller and Engstrom, 2007*) suggests the possibility of clustered vocal behavior in some mouse strains not included in our data. Second, it is possible that the difference in clustered and nonclustered data depends crucially on dataset size. If real syllables even occasionally fall between well-defined clusters, a large enough dataset might lightly 'fill in' true gaps. Conversely, even highly clustered data may look more or less continuous given an insufficient number of samples per cluster. While this is not likely given the more than 15,000 syllables in *Figure 5*, it is difficult to rule out in general. Finally, our purely signal-level analysis of vocal behavior cannot address the possibility that a continuous distribution of syllables could nevertheless be perceived categorically. For example, swamp sparrows exhibit categorical neural and behavioral responses to changes in syllable duration (*Prather et al., 2009*). Nonetheless, we argue that, without empirical evidence to this effect in rodents, caution is in order when interpreting the apparent continuum of USV syllables in categorical terms.

Lastly, we showed how a 'shotgun VAE' approach can be used to extend our approach to the quantification of moment-by-moment vocal variability. In previous studies, syllable variability has only been quantified for certain well-characterized syllables like harmonic stacks in zebra finch song (*Kao and Brainard, 2006*). Our method, by contrast, provides a continuous variability measure for all syllables (*Figure 6c*). This is particularly useful for studies of the neural basis of this vocal variability, which is hypothesized to operate on millisecond to tens-of-milliseconds timescales (*Fee and Goldberg, 2011*).

Nonetheless, as a data-driven method, our approach carries some drawbacks. Most notably, the VAE must be trained on a per-dataset basis. This is more computationally intensive than calculating typical acoustic features (≈1 hr training times on a GPU) and also prevents direct comparisons across datasets unless they are trained together in a single model. Additionally, the resulting learned features, representing nonlinear, nonseparable acoustic effects, are somewhat less interpretable than named acoustic features like duration and spectral entropy. However, several recent studies in the VAE literature have attempted to address this issue by focusing on the introduction of covariates (*Sohn et al., 2015*; *Louizos et al., 2015*; *Khemakhem et al., 2019*) and 'disentangling' approaches that attempt to learn independent sources of variation in the data (*Higgins et al., 2017*; *Burgess et al., 2018*), which we consider to be promising future directions. We also note that great progress in generating raw audio has been made in the past few years, potentially enabling similar approaches that bypass an intermediate spectrogram representation (*Avd et al., 2016*; *Kong et al., 2020*).

Finally, we note that while our focus in this work is vocal behavior, our training data are simply syllable spectrogram images. Similar VAE approaches could also be applied to other kinds of data summarizable as images or vectors. The shotgun VAE approach could likewise be applied to sequences of such vectors, potentially revealing structures like those in *Figure 6b*. More broadly, our results suggest that data-driven dimensionality reduction methods, particularly modern nonlinear, overparameterized

methods, and the latent spaces that come with them, offer a promising avenue for the study of many types of complex behavior.

## Materials and methods

### Animal statement

All experiments were conducted according to protocols approved by the Duke University Institutional Animal Care and Use Committee (mice: A171-20-08, birds: A172-20-08).

### Recordings

Recordings of C57BL/6 and DBA/2 mice were obtained from the MUPET Wiki (*Van Segbroeck M et al., 2019*). These recordings are used in *Figure 2*, *Figure 2—figure supplements 1–5*, *Figure 3*, *Figure 3—figure supplements 1–3*, *Figure 4d–e*, *Figure 4—figure supplement 2*, *Figure 5e*, and Appendix 1. Low-dimensional learned feature spaces quantify individual and group differences in vocal repertoires (*Appendix 1—table 2*).

Additional recordings of female-directed mouse USVs are used in *Figure 4* and *Figure 4—figure supplement 3*. These recordings comprise 36 male mice from various genetic backgrounds over 118 recording sessions of roughly 20 minutes each (≈40 total hours, 156,180 [*Appendix 1—table 1*] total syllables). USVs were recorded with an ultrasonic microphone (Avisoft, CMPA/CM16), amplified (Presonus TubePreV2), and digitized at 300 kHz (Spike 7, CED). A subset of these recordings corresponding to a single individual (17,400 syllables) is further used in *Figures 5b–d and 6a*, *Figure 2—figure supplement 6*, *Figure 5—figure supplement 1*, *Figure 5—figure supplement 2*, *Figure 5—figure supplement 4*, and *Figure 6—figure supplement 1*. Because these recordings contained more noise than the first set of C57/DBA recordings, we removed false-positive syllables by training the VAE on all detected syllables, projecting latent syllables to two dimensions, and then removing syllables contained within the resulting cluster of noise syllables, with 15,712 syllables remaining (see *Figure 2—figure supplement 6*).

A single male zebra finch was recorded over a 2-day period (153–154 days post-hatch [dph]) in both female-directed and undirected contexts (14,270 total syllables, 1100 directed). Song was recorded with Sound Analysis Pro 2011.104 (*Tchernichovski and Mitra, 2004*) in a soundproof box. These recordings are used in *Figure 2*, *Figure 2—figure supplement 1*, *Figure 2—figure supplement 2*, *Figure 2—figure supplement 3*, *Figure 2—figure supplement 4*, *Figure 2—figure supplement 5*, *Figure 3—figure supplement 2*, *Figure 4*, *Figure 4—figure supplement 1*, *Figure 5*, *Figure 5—figure supplement 1*, *Figure 5—figure supplement 3*, *Figure 5—figure supplement 4*, *Figure 5—figure supplement 5*, *Figure 6*, *Figure 6—figure supplement 1*, *Figure 6—figure supplement 2*, and *Appendix 1—table 1*.

For *Figure 7*, we selected 10 adult, normally reared birds from different breeding cages in our colony. Until at least 60 dph, each of these pupil birds had interacted with only one adult male, the tutor from his respective breeding cage. We recorded the adult (>90 dph) vocalizations of these pupil birds for 5–12 days each with Sound Analysis Pro 2011.104 (*Tchernichovski and Mitra, 2004*), then recorded their respective tutors under the same conditions for 5–12 days each. These recordings are used in *Figure 7* and *Figure 7—figure supplement 1*.

### Software comparisons

We compared our VAE method to three widely used vocal analysis packages: MUPET 2.0 (*Van Segbroeck et al., 2017*), DeepSqueak 2.0 (*Coffey et al., 2019*) (for mouse USVs), and SAP 2011.104 (*Tchernichovski and Mitra, 2004*) (for birdsong), each with default parameter settings. MUPET clusters were found using the minimum number of clusters (10). DeepSqueak features were generated using the DeepSqueak 'import from MUPET' feature.

#### Audio segmenting

For all mouse USV datasets, individual syllable onsets and offsets were detected using MUPET with default parameter settings. The shotgun VAE analysis in *Figure 6* was restricted to manually defined regions (bouts) of vocalization. In this figure, zebra finch songs were segmented semi-automatically: first, we selected four representative song motifs from each individual. Then we converted these to

spectrograms using a Short Time Fourier Transform with Hann windows of length 512 and overlap of 256, averaged these spectrograms, and blurred the result using a Gaussian filter with 0.5 pixel standard deviation. The result was a song template used to match against the remaining data. Specifically, we looked for local maxima in the normalized cross-correlation between the template and each audio file. Matches corresponded to local maxima with cross-correlations above 1.8 median absolute deviations from the median, calculated on a per-audio-file basis. A spectrogram was then computed for each match. All match spectrograms were then projected to two dimensions using UMAP (*McInnes et al., 2018*), from which a single well-defined cluster, containing stereotyped song, was retained. Zebra finch syllable onsets and offsets were then detected using SAP (*Tchernichovski and Mitra, 2004*) on this collection of song renditions using default parameter settings. After segmentation, syllable spectrograms were projected to two dimensions using UMAP, and eight well-defined clusters of incorrectly segmented syllables were removed, leaving six well-defined clusters of song syllables. For *Figure 7*, song motifs were hand-labeled for approximately 10 min of song-rich audio per animal. These labels were used to train an automated segmentation tool, *vak* 0.3.1 (*Nicholson and Cohen, 2020*), for each animal. Trained *vak* models were used to automatically label motifs in the remaining audio data for each animal. Automatic segmentation sometimes divided single motifs or joined multiple motifs. To correct for these errors, short (<50 ms) gaps inside motifs were eliminated. After this correction, putative motif segments with durations outside 0.4–1.5 s were discarded. Syllables segments were derived from a subset of the *vak* motif segments by aligning the motif amplitude traces and manually determining syllable boundaries, resulting in 75,430 total syllable segments.

## Spectrograms

Spectrograms were computed using the log modulus of a signal's Short Time Fourier Transform, computed using Hann windows of length 512 and overlap of 256 for bird vocalization, and length 1024 and overlap 512 for mouse vocalization. Sample rates were 32 kHz for bird vocalization and 250 kHz for mouse vocalization, except for the recordings in *Figure 3f*, which were sampled at a rate of 300 kHz. The resulting time/frequency representation was then interpolated at desired frequencies and times. Frequencies were mel-spaced from 0.4 to 8 kHz for bird recordings in *Figure 7*, mel-spaced from 0.4 to 10 kHz for all other bird recordings, and linearly spaced from 30 to 110 kHz for mouse recordings. For both species, syllables longer than $t_{\max} = 200\mathrm{ms}$ were discarded. Additionally, short syllables were stretched in time in a way that preserved relative duration, but encouraged the VAE to represent fine temporal details. Specifically, a syllable of duration $t$ was stretched by a factor of $\sqrt{\frac{t_{\max}}{t}}$. See *Figure 2—figure supplement 5* for a comparison to linear frequency spacing for zebra finches and no time stretching for mice and zebra finches. The resulting spectrograms were then clipped to manually tuned minimum and maximum values. The values were then linearly stretched to lie in the interval [0,1]. The resulting spectrograms were 128 × 128 = 16,384 pixels, with syllables shorter than $t_{\max}$ zero-padded symmetrically.

## Model training

Our VAE is implemented in PyTorch (v1.1.0) and trained to maximize the standard evidence lower bound (ELBO) objective using the reparameterization trick and ADAM optimization (*Kingma and Welling, 2013*; *Rezende et al., 2014*; *Paszke et al., 2017*; *Kingma and Ba, 2014*). The encoder and decoder are deep convolutional neural networks with fixed architecture diagrammed in *Figure 1—figure supplement 1*. The latent dimension was fixed to 32, which was found to be sufficient for all training runs. The approximate posterior was parameterized as a normal distribution with low rank plus diagonal covariance: $q(z) = \mathcal{N}(z; \mu, uu^\top + \mathrm{diag}(d))$, where μ is the latent mean, u is a 32 × 1 covariance factor, and $d$ was the latent diagonal, a vector of length 32. The observation distribution was parameterized as $\mathcal{N}(\mu, 0.1I)$, where μ was the output of the decoder. All activation functions were Rectified Linear Units. Learning rate was set to $10^{-3}$ and batch size was set to 64.

## Comparison of VAE and handpicked features

For each analysis tool (MUPET, DeepSqueak, SAP), we assembled two feature sets: one calculated by the comparison tool (e.g., MUPET features) and one a matched VAE set. For the first set, each feature calculated by the program was z-scored and all components with nonzero variance were retained (9/9, 10/10, and 13/14 components for MUPET, DeepSqueak, and SAP, respectively). For the second

set, we trained a VAE on all syllables, computed latent means of these via the VAE encoder, and removed principal components containing less than 1% of the total feature variance (7, 5, and 5 out of 32 components retained for MUPET, DeepSqueak, and SAP syllables, respectively). Each feature set was used as a basis for predicting the features in the other set using $k$-nearest neighbors regression with $k$ set to 10 and nearest neighbors determined using Euclidean distance in the assembled feature spaces. The variance-explained value reported is the average over five shuffled train/test folds (*Figure 2e*).

Unlike latent features, traditional features do not come equipped with a natural scaling. For this reason, we z-scored traditional features to avoid tethering our analyses to the identities of particular acoustic features involved. Then, to fairly compare the effective dimensionalities of traditional and acoustic features in *Figure 2d*, we thus also z-scored the latent features as well, thereby disregarding the natural scaling of the latent features. PCA was then performed on the resulting scaled feature set.

## Birdsong variability index

For *Figure 4c*, given a set $\{z_i | i = 1 \ldots n\}$ of feature vectors of $n$ syllables, we defined a variability index for the data as follows:

$$\text{V.I.} = \min_{z_i} \rho(z_i) \tag{1}$$

where $\rho(z)$ is proportional to a robust estimator of the variance of the data around $z$:

$$\rho(z) = \operatorname*{median}_{z_j} \|z - z_j\|_2^2 \tag{2}$$

We calculate the above metric for every combination of syllable (A–F), feature set (SAP-generated vs. VAE-generated), and social context (directed vs. undirected) and report the variability index of the directed condition relative to the variability index of the undirected condition (*Figure 4c*).

For *Figure 6e*, we would ideally use the variability index defined above, but $\rho(z)$ is expensive to compute for each datapoint, as required in (1). Thus, we use an approximate center point defined by the median along each *coordinate*: $\hat{z}^i \equiv \operatorname{median}(z^i)$, where the superscript here represents the ith coordinate of $z$. That is, $\hat{z}$ contains the medians of the marginal distributions. This value is calculated for each combination of time point and social context (directed vs. undirected) and plotted in *Figure 6e*.

## Maximum mean discrepancy

We used the MMD integral probability metric to quantify differences in sets of syllables (*Gretton et al., 2012*). Given random variables $x$ and $y$, MMD is defined over a function class $\mathcal{F}$ as $\sup_{f \in \mathcal{F}} \mathbb{E}_x[f(x)] - \mathbb{E}_y[f(y)]$. Here, $\mathcal{F}$ was taken to the set of functions on the unit ball in a reproducing kernel Hilbert space with fixed spherical Gaussian kernel. For *Figure 4e–f*, the kernel width $\sigma$ was chosen to be the median distance between points in the aggregate sample, a common heuristic (*Gretton et al., 2012*). In *Figure 7*, the kernel bandwidth was chosen to be 25% of the median distance between points in the aggregate sample in order to focus on finer differences in distributions. For *Figure 4e*, we obtained 20 approximately 6.5 min recordings of male C57BL/6 mice and 20 approximately 6.5 min recordings of male DBA/2 mice (see Recordings). Latent means of USVs from a single recording were treated as independent and identically distributed draws from a recording-specific USV distribution, and MMD was estimated using these latent means. In *Figure 4e*, the MMD values are plotted as a matrix and the order of rows was obtained by agglomerative clustering. In *Figure 4f* and *Figure 7*, the same procedure was followed. For *Figure 4f*, a t-SNE was computed for each recording session, with the distance between recording sessions taken to be the estimated MMD between them (see *Figure 4—figure supplement 3* for the MMD matrix).

## Unsupervised clustering metrics

We used three unsupervised clustering metrics to assess the quality of clustering for both zebra finch and mouse syllables: the mean silhouette coefficient (*Rousseeuw, 1987*), the Calinski–Harabasz Index (*Caliński and Harabasz, 1974*), and the Davies–Bouldin Index (*Davies and Bouldin, 1979*). For each species (zebra finch and mouse), we partitioned the data for 10-fold cross-validation (train on 9/10, test on 1/10 held out). For a null comparison, for each of 10% disjoint subsets of the data, we created

a synthetic Gaussian noise dataset matched for covariance and number of samples. These synthetic noise datasets were then used to produce the dotted line in *Figure 5d*.

For each data split, we clustered using a GMM with full covariance using Expectation Maximization on the training set. We then evaluated each clustering metric on the test set. The number of clusters, $k$, was set to six in *Figure 5d*, but qualitatively similar results were obtained when $k$ was allowed to vary between 2 and 12 (*Figure 5—figure supplement 1*). Reported values in *Figure 5d* and *Figure 5—figure supplement 1* are the differences in unsupervised metrics on real data and Gaussian noise for each cross-validation fold, with a possible sign change to indicate higher values as more clustered.

## Shotgun VAE

To perform the analysis in *Figure 6a–b*, regions of active vocalization were defined manually for both species (22 min of mouse recordings, 2 min of zebra finch recordings). Zebra finch bouts containing only calls and no song motifs were excluded. For both species, the duration of audio chunks was chosen to be roughly as long as the longest syllables (zebra finch: 120 ms; mouse: 200 ms). No explicit training set was made. Rather, onsets and offsets were drawn uniformly at random from the set of fixed-duration segments and the corresponding spectrograms were computed on a per-datapoint basis. Thus, the VAE likely never encountered the same spectrogram twice, encouraging it to learn the underlying time series.

To perform the variability reduction analysis in *Figure 6d–e*, song renditions were collected (see Audio segmenting) and a spectrogram was computed for each. The whole collection of spectrograms was then jointly warped using piecewise-linear time warping (*Williams et al., 2019*). Fixed-duration training spectrograms were made by interpolating normal spectrograms (as described in Spectrograms) at linearly spaced time points in warped time, generally corresponding to nonlinearly spaced time points in real time. As above, spectrograms were made during training on a per-datapoint basis. After training the VAE on these spectrograms, latent means were collected for 200 spectrograms for each song rendition, linearly spaced in warped time from the beginning to the end of the song bout. Lastly, for each combination of condition (directed vs. undirected song) and time point, the variability index described above was calculated. A total of 186 directed and 2227 undirected song renditions were collected and analyzed.

To generate the shotgun VAE training set for *Figure 7*, 2000 *vak*-labeled motifs were selected from each animal (see Audio segmenting). A single 60 ms window was drawn from each motif to create a training set of 40,000 total spectrogram windows drawn from the 20-animal cohort. After training a VAE on this dataset, the hand-labeled motif segments used to train *vak* models (see Audio segmenting) were segmented into overlapping 60 ms windows that spanned each motif with an 8 ms step size between successive windows (52,826 total windows). These windows were reduced with the trained VAE and their latent means subsequently analyzed.

## Modified UMAP

Although the standard UMAP embedding of shotgun VAE latents from single-finch datasets generates points along smoothly varying strands (see *Figure 6b*), UMAP typically broke motifs into multiple strand-like pieces in the 20-animal dataset from *Figure 7*. To encourage embeddings that preserve the neighbor relationship of successive windows, we modified the distance measure underlying the UMAP. First, we computed the complete pairwise Euclidean distance matrix between all windows in latent space. Then, we artificially decreased the distance between successive windows from the same motif by multiplying corresponding distance matrix entries by $10^{-3}$. This precomputed distance matrix was then passed to UMAP as a parameter. See *Figure 7—figure supplement 1* for a comparison of the two UMAP projections.

## Data and code availability statement

The latest version of Autoencoded Vocal Analysis, the Python package used to generate, plot, and analyze latent features, is available online: https://github.com/pearsonlab/autoencoded-vocal-analysis (*Goffinet, 2021*; copy archived at swh:1:rev:f512adcae3f4c5795558e2131e54c36daf23b904). Mouse and zebra finch recordings used in this study are archived on the Duke Digital Repository: https://doi.org/10.7924/r4gq6zn8w.

## Acknowledgements

This work was supported by NIH grants R01-DC013826 (RM, JP), R01-NS118424 (RM, JP), R01-NS099288 (RM), R01-MH117778 (RM), and F31-HD098772 (SB) and by a hardware grant from the NVIDIA corporation.

## Additional information

### Funding

| Funder | Grant reference number | Author |
| --- | --- | --- |
| National Institute of Mental Health | R01-MH117778 | Richard Mooney |
| National Institute of Neurological Disorders and Stroke | R01-NS118424 | Richard Mooney John Pearson |
| National Institute on Deafness and Other Communication Disorders | R01-DC013826 | Richard Mooney John Pearson |
| National Institute of Neurological Disorders and Stroke | R01-NS099288 | Richard Mooney |
| Eunice Kennedy Shriver National Institute of Child Health and Human Development | F31-HD098772 | Samuel Brudner |

The funders had no role in study design, data collection and interpretation, or the decision to submit the work for publication.

### Author contributions

Jack Goffinet, Conceptualization, Formal analysis, Investigation, Methodology, software, validation, Visualization, Writing – original draft, Writing – review and editing; Samuel Brudner, Formal analysis, Investigation, Methodology, validation, Visualization, Writing – review and editing; Richard Mooney, Conceptualization, Funding acquisition, Investigation, Project administration, resources, Supervision, Writing – original draft, Writing – review and editing; John Pearson, Conceptualization, Formal analysis, Funding acquisition, Investigation, Methodology, Project administration, Supervision, Visualization, Writing – original draft, Writing – review and editing

### Author ORCIDs

Jack Goffinet http://orcid.org/0000-0001-6729-0848
Samuel Brudner http://orcid.org/0000-0002-6043-9328
Richard Mooney http://orcid.org/0000-0002-3308-1367
John Pearson http://orcid.org/0000-0002-9876-7837

### Ethics

All data generated in conjunction for this study were generated by experiments performed in strict accordance with the recommendations in the Guide for the Care and Use of Laboratory Animals of the National Institutes of Health. All of the animals were handled according to approved institutional animal care and use committee (IACUC) protocols of Duke University, protocol numbers A171-20-08 and A172-20-08.

### Decision letter and Author response

Decision letter https://doi.org/10.7554/eLife.67855.sa1
Author response https://doi.org/10.7554/eLife.67855.sa2

## Additional files

### Supplementary files
• Transparent reporting form

### Data availability
Dataset 1: Online, publicly available MUPET dataset: ~5GB Available at: https://github.com/mvan-segbroeck/mupet/wiki/MUPET-wiki Figs: 2, 3, 4d-e.

Dataset 2: Single zebra finch data: ~200-400 MB of audio generated as part of work in progress in Mooney Lab. Figs: 2e-f, 4a-c, 5a, 5d, 6b-e.

Dataset 3: Mouse USV dataset: ~30-40 GB of audio generated as part of work in progress in Mooney Lab. Figs: 4f.

Dataset 5: This is a subset of dataset 3, taken from a single mouse: ~1GB of audio. Figs: 5b-e, 6a.

Dataset 6: 10 zebra finch pupil/tutor pairs: ~60 GB of audio generated as part of work in progress in Mooney Lab. Figs: 7.

Datasets 2-6 are archived in the Duke Digital Repository (https://doi.org/10.7924/r4gq6zn8w).

The following previously published datasets were used:

| Author(s) | Year | Dataset title | Dataset URL | Database and Identifier |
|---|---|---|---|---|
| Pearson J, Mooney R, Brudner S, Goffinet J | 2021 | | https://doi.org/10.7924/r4gq6zn8w | Duke Digital Repository, 10.7924/r4gq6zn8w |

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

## Appendix 1

**Appendix 1—table 1.** Comparison of feature sets on the downstream task of predicting finch social context.

(directed vs. undirected context) given acoustic features of single syllables.Classification accuracy, in percent, averaged over five disjoint, class-balanced splits of the data is reported. Empirical standard deviation is shown in parentheses. Euclidean distance is used for nearest neighbor classifiers. Each SAP acoustic feature is independently z-scored as a preprocessing step. Latent feature dimension is truncated when >99% of the feature variance is explained. Random forest (RF) classifiers use 100 trees and the Gini impurity criterion. The multilayer perceptron (MLP) classifiers are two-layer networks with a hidden layer size of 100, ReLU activations, and an L2 weight regularization parameter 'alpha,' trained with ADAM optimization with a learning rate of $10^{-3}$ for 200 epochs. D denotes the dimension of each feature set, with Gaussian random projections used to decrease the dimension of spectrograms.

**Predicting finch social context (Figure 4a–c)**

| | Spectrogram | | | SAP | Latent |
|---|---|---|---|---|---|
| | D = 10 | D = 30 | D = 100 | D = 13 | D = 5 |
| $k$-NN ($k = 3$) | 92.5 (0.2) | 95.3 (0.1) | **97.3 (0.3)** | 93.0 (0.3) | 96.9 (0.2) |
| $k$-NN ($k = 10$) | 93.0 (0.2) | 95.3 (0.2) | 97.1 (0.3) | 93.2 (0.1) | 96.7 (0.2) |
| $k$-NN ($k = 30$) | 92.7 (0.1) | 94.2 (0.2) | 96.0 (0.2) | 92.8 (0.1) | 96.3 (0.1) |
| RF (depth = 10) | 92.6 (0.1) | 92.7 (0.1) | 93.1 (0.1) | 92.8 (0.1) | 94.9 (0.2) |
| RF (depth = 15) | 92.7 (0.1) | 93.2 (0.2) | 93.6 (0.1) | 93.6 (0.2) | 96.1 (0.1) |
| RF (depth = 20) | 92.8 (0.1) | 93.4 (0.1) | 93.8 (0.1) | 93.8 (0.2) | 96.4 (0.2) |
| MLP ($\alpha = 0.1$) | 92.8 (0.4) | 95.4 (0.4) | **97.6 (0.3)** | 92.9 (0.1) | 95.7 (0.1) |
| MLP ($\alpha = 0.01$) | 92.9 (0.3) | 95.4 (0.3) | **97.5 (0.2)** | 93.1 (0.2) | 96.2 (0.1) |
| MLP ($\alpha = 0.001$) | 92.7 (0.6) | 95.2 (0.5) | **97.5 (0.2)** | 93.0 (0.2) | 96.3 (0.0) |

**Appendix 1—table 2.** Comparison of feature sets on the downstream task of predicting mouse strain.

(C57 vs.DBA) given acoustic features of single syllables. Classification accuracy, in percent, averaged over five disjoint, class-balanced splits of the data is reported. Empirical standard deviation is shown in parentheses. Euclidean distance is used for nearest neighbor classifiers. Each MUPET and DeepSqueak acoustic feature is independently z-scored as a preprocessing step. Latent features dimension is truncated when >99% of the feature variance is explained. Random forest (RF) classifiers use 100 trees and the Gini impurity criterion. The multilayer perceptron (MLP) classifiers are two-layer networks with a hidden layer size of 100, ReLU activations, and an L2 weight regularization parameter 'alpha,' trained with ADAM optimization with a learning rate of $10^{-3}$ for 200 epochs. D denotes the dimension of each feature set, with Gaussian random projections used to decrease the dimension of spectrograms.

**Predicting mouse strain (Figure 4d–e)**

| | Spectrogram | | | MUPET | DeepSqueak | Latent |
|---|---|---|---|---|---|---|
| | D = 10 | D = 30 | D = 100 | D = 9 | D = 10 | D = 7 |
| $k$-NN ($k = 3$) | 68.1 (0.2) | 76.4 (0.3) | 82.3 (0.5) | 86.1 (0.2) | 79.0 (0.3) | 89.8 (0.2) |
| $k$-NN ($k = 10$) | 71.0 (0.3) | 78.2 (0.1) | 82.7 (0.6) | 87.0 (0.1) | 80.7 (0.3) | 90.7 (0.4) |
| $k$-NN ($k = 30$) | 72.8 (0.3) | 78.5 (0.2) | 81.3 (0.5) | 86.8 (0.2) | 81.0 (0.2) | 90.3 (0.4) |

*Appendix 1—table 2 Continued on next page*

*Appendix 1—table 2 Continued*

**Predicting mouse strain (Figure 4d–e)**

|  | Spectrogram | | | MUPET | DeepSqueak | Latent |
|---|---|---|---|---|---|---|
|  | D = 10 | D = 30 | D = 100 | D = 9 | D = 10 | D = 7 |
| RF (depth = 10) | 72.8 (0.2) | 76.6 (0.2) | 79.1 (0.3) | 87.4 (0.5) | 81.2 (0.4) | 88.1 (0.5) |
| RF (depth = 15) | 73.1 (0.3) | 78.0 (0.3) | 80.5 (0.2) | 87.9 (0.4) | 82.1 (0.3) | 89.6 (0.4) |
| RF (depth = 20) | 73.2 (0.2) | 78.3 (0.2) | 80.7 (0.3) | 87.9 (0.4) | 81.9 (0.3) | 89.6 (0.4) |
| MLP ($\alpha$ = 0.1) | 72.4 (0.3) | 79.1 (0.4) | 84.5 (0.3) | 87.8 (0.2) | 82.1 (0.4) | **90.1 (0.3)** |
| MLP ($\alpha$ = 0.01) | 72.3 (0.4) | 78.6 (0.3) | 82.9 (0.4) | 88.1 (0.3) | 82.4 (0.4) | **90.0 (0.4)** |
| MLP ($\alpha$ = 0.001) | 72.4 (0.4) | 78.5 (0.8) | 82.8 (0.1) | 87.9 (0.2) | 82.4 (0.3) | **90.4 (0.3)** |

**Appendix 1—table 3.** Comparison of feature sets on the downstream task of predicting mouse identity given acoustic features of single syllables.

Classification accuracy, in percent, averaged over five disjoint, class-balanced splits of the data is reported. A class-weighted log-likelihood loss is targeted to help correct for class imbalance. Empirical standard deviation is shown in parentheses. Each MUPET acoustic feature is independently z-scored as a preprocessing step. Latent feature principal components are truncated when >99% of the feature variance is explained. The multilayer perceptron (MLP) classifiers are two-layer networks with a hidden layer size of 100, ReLU activations, and an L2 weight regularization parameter 'alpha,' trained with ADAM optimization with a learning rate of $10^{-3}$ for 200 epochs. Chance performance is 2.8% for top-1 accuracy and 13.9% for top-5 accuracy. D denotes the dimension of each feature set, with Gaussian random projections used to decrease the dimension of spectrograms.

**Predicting mouse identity (Figure 4f)**

|  | Spectrogram | | | MUPET | Latent |
|---|---|---|---|---|---|
|  | D = 10 | D = 30 | D = 100 | D = 9 | D = 8 |
| *Top-1 accuracy* | | | | | |
| MLP ($\alpha$ = 0.01) | 9.9 (0.2) | 14.9 (0.2) | 20.4 (0.4) | 14.7 (0.2) | 17.0 (0.3) |
| MLP ($\alpha$ = 0.001) | 10.8 (0.1) | 17.3 (0.4) | **25.3 (0.3)** | 19.0 (0.3) | 22.7 (0.5) |
| MLP ($\alpha$ = 0.0001) | 10.7 (0.2) | 17.3 (0.3) | **25.1 (0.3)** | 20.6 (0.4) | 24.0 (0.2) |
| *Top-5 accuracy* | | | | | |
| MLP ($\alpha$ = 0.01) | 36.6 (0.4) | 45.1 (0.5) | 55.0 (0.3) | 46.5 (0.3) | 49.9 (0.4) |
| MLP ($\alpha$ = 0.001) | 38.6 (0.2) | 50.7 (0.6) | **62.9 (0.4)** | 54.0 (0.2) | 59.2 (0.6) |
| MLP ($\alpha$ = 0.0001) | 38.7 (0.5) | 50.8 (0.3) | **63.2 (0.4)** | 57.3 (0.4) | 61.6 (0.4) |

