## [Decision Letter]

**Acceptance summary:**

Animal vocalizations are notoriously complex and difficult to categorize. Traditionally, sounds are transformed into spectrograms, which are then segmented into syllables and analyzed according to hand-selected features such as pitch, amplitude and frequency modulation. Here, the authors take a new approach: they use variational autoencoders to analyze vocalizations from songbirds and mice and find that they can quantify the similarity between distinct utterances. This approach will complement existing sound analysis methods to further our understanding of animal social behavior.

**Decision letter after peer review:**

[Editors’ note: the authors submitted for reconsideration following the decision after peer review. What follows is the decision letter after the first round of review.]

Thank you for submitting your work entitled "Low-dimensional learned feature spaces quantify individual and group differences in vocal repertoires" for consideration by *eLife*. Your article has been reviewed by 4 peer reviewers, including Jesse H Goldberg as the Reviewing Editor and Reviewer #1, and the evaluation has been overseen by a Senior Editor. The following individuals involved in review of your submission have agreed to reveal their identity: Ofer Tchernichovski (Reviewer #3); Scott W Linderman (Reviewer #4).

Our decision has been reached after consultation between the reviewers. Based on these discussions and the individual reviews below, we regret to inform you that your work will not be considered further for publication in *eLife* (but note caveat below).

The reviewers mostly agreed that the VAE is a potentially interesting approach to categorizing vocalization data, and there was enthusiasm about the codebase available in github. Some major problems that arose in review were (1) lack of strong behavioral insights; (2) lack of clarity about data pre-processing – and how this would affect results; and (3) concern about novelty given the widespread use of VAEs for similar problems.

We would, in principle, be open to considering a revised version of this manuscript if the relatively long list of concerns by reviewers 2 and 4 were adequately addressed and if the VAE approach could perform similarity-score metrics (as requested by reviewer 3).

*Reviewer #1:*

The authors proposed the use of variational autoencoder (VAE) to vocal recordings of model organisms (mouse and zebra finch) to better capture the acoustic features that are missed by conventional acoustic metrics. This manuscript explores the effectiveness of the VAE approach from two perspectives: (i) a motifs based clustering which seeks to match the acoustics data against several predetermined template and (ii) an unsupervised clustering based on randomly segmented vocal recordings. These approaches involve the generation of a collection of images from spectrograms that are then fed to variational encoders to estimate the number of latent variables. With these latent variables, the authors then employed UMAP to visualize the variation within the dataset. The analyses are well conducted and will be useful for broad range of scientists investigating animal vocalizations.

i. From the zebra finch's discussion, this approach performs well in clustering the song syllables based on the four motifs predefined and at the same time could delineate the differences between directed and undirected songs as compared to previous acoustic metrics. While the authors provided a comparison between the ability of SAP acoustics features and VAE latent features in differentiating the directed and undirected birdsong, a comparison between the clustering of song syllables with the use of SAP features and VAE latent feature was not offered (Figure 5a). It would be interesting to see a side-by-side comparison of the 'Goodness of Clustering' metric in this VAE approach vs SAP.

ii. As for the randomly segmented vocal recordings, this method could generate a manifold where different acoustic features were situated in different regions and offer a continuous variability measure for all syllables. It is worth noting that the vocal recordings were randomly segmented based on the length of one syllable. What happens when the length is extended beyond that? Will the manifold produced look like the one shown in Figure 6?

iii. As for the mouse vocal recordings, the VAE approach generates a continuum-like points cloud that performs reasonably well in differentiating different acoustic features of the mouse vocalizations albeit the lack of boundaries in separating them. Could the smooth variation of points be due to the sampling rate? The mouse vocalizations were sampled at a much higher rate (10x) as compared to bird vocalizations. I would expect a greater resolution for the mouse data and thus the VAE can capture more subtle differences between the vocalizations, yielding a continuum-like points cloud. Of course these sampling rate differences are justified because of the different spectral properties of birdsong and mouse USVs – but a simple sentence or two about how sampling rates may affect these analyses would be useful..

*Reviewer #2:*

Authors introduce a new method for behavioral analysis of vocalizations that supposedly improves on weaknesses of handpicked features, which is that they miss variability and introduce undesirable correlations. They apply VAEs to bird and mouse vocalizations and perform analysis of the latent space. They show that VAEs outperform handpicked features on some simple analysis tasks including time-resolved analysis of variability.

Overall, even if it is correct that their method outperforms traditional-features based analysis, I don't think this insight is in any way relevant. Essentially, it is like saying: here is a bad way of doing things, and we offer one that is slightly better but still much worse than the gold standard. The problem is that there are many methods out there for doing things right from the start, so I cannot see a need for VAEs for the problems addressed. The problem with VAEs is that they are compressive (not representing all aspects of the data), which is why they are less desirable than methods that operate on equivalent representations of the raw data. If you are given money to buy groceries, why would you throw away part of it before you enter the store?

Overall it seems authors have not much to argue for other than their opposition to handpicked features. They pretend not much is out there other than these features for behavioral analysis, which does not reflect the wealth of methods out there. They have not even given appropriate credit to the real usefulness of the handpicked-feature approach, which is interpretability (which their method lacks). For example, one main utility of handpicked features is that some are learned. Songbirds have been shown to learn pitch from a tutor, so pitch is relevant. The same cannot be said about their latent features: which one of these do birds learn from a tutor?

Also, correlations among features that authors criticize as a bug can be seen as a feature, it provides insights into the aspects of their vocalizations that animals cannot control independently.

I could also not detect any new behavioral insights. The only real strength in their manuscript I could detect is that their method allows them to visually segregate directed from undirected song (Figure 4b), vocalizations from different mice strains (Figure 4f), and songbird syllables (Figure 5a). Thus, their method could be a suitable pre-processing for clustering efforts. Also, on a positive note, their method also produces some interesting looking plots (Figure 6b) of to-be identified utility.

L15 Authors claim that finding concise metrics remains a challenge, despite the large number of concise metrics ranging from Euclidean, Hamming, cosine, Wasserstein, Jensen-Shannon, Bhattacharia, Levenstein, ROUGE, etc. Surprisingly, their paper deals with the least concise metric imaginable, a deep autoencoder with thousands of parameters!

L17 Also not clear what would substantiate the claim that vocal behavior remains poorly understood.

L19 They mention several powerful approaches to enable automatic analysis of vocalizations but cite none.

L20 Given the wished-for generality of the problem they would like to study, it sounds strange that key to a successful approach would be a software package. Software packages are the last stage of analysis tool development.

L26 [10] did not discover overnight consolidation of learned birdsong as claimed, but instead 'inappropriate consolidation', which is challenged by a more recent analysis in [26].

L29 Authors criticize 'correlations among features', claiming these could result in redundant characterizations of vocalizations. Implicitly they argue for features that do not correlate, e.g. independent component analysis. But again, no citations, no following up on the embarked ideas. Correlations could actually be a feature of the behavior itself (and quite interesting to study).

I do not believe that mouse USV syllables form a continuum of syllables. Because authors did not find clusters, this does not mean that they are not there. Rather than trying to find shortcomings (e.g. of their method or the number of samples analyzed), authors generalize from a negative finding to inexistence. By their rejection of vocal clustering, they also ignore previous results showing such clustering [18, 4, 48, 6, 16]. Quite audacious. Is the moon still there when I close my eyes?

In the caption of Figure 2d, authors state 'an ideal representation would exhibit minimal off-diagonal correlation', i.e., ICA is ideal. Why do we need VAEs then if ICA is ideal?

Caption Figure 2e, the representational capacity will depend on the number of features, which is not reported. Same for Figure 2f, the more features used, the more principal components will be needed, so this may be a trivial effect of unequal number of features.

With regards to Figure 2f, it is not even clear from their analysis whether for a given fixed dimensionality, VAEs encode more variance than simple PCA, and if so, at what cost on memory (principal components vs auto-encoder network). For example, in the original Hinton paper in Science, the outcome of this analysis was rather surprising (VAEs are not clearly outperforming PCA in terms of representational capacity).

Last paragraph of Page 4, other than some pretty pictures (Figure S4) there is no (numerical) evidence for their claims of superiority of their latent features.

L115-125 and Figure 4: This is an ill-advised analysis. Why would one choose SAP features to detect changes in song? It is obvious that change detection requires the most sensitive analysis possible, so why would one perform compression beforehand? Same goes for their latent features. Even if it is better than SAP, why would one choose it and not the raw data itself (and a suitable standard metric)?

Same for L 126-L135 on data in mice.

L 173-L187: Authors argue about clustering failures of MUPET using UMAP representations, ignoring the fact that UMAP provides a faulty account of true distance. Their analysis of latent features is a step in the right direction, but falls short of analysis of the raw data (or an equivalent representation, e.g. https://asa.scitation.org/doi/abs/10.1121/1.4731466 and others).

L214: Their method is not useful for analyzing moment-by-moment variability of song, because they need to pre-process songs by 'warping each in time to account for well-documented differences in rhythm and tempo', which is the only problem that would complicate a more rigorous analysis.*Reviewer #3:*

This manuscript presents a new implementation of variational autoencoder machine learning algorithm (VAE) for the analysis of animal vocalization. The new method is impressive and powerful compared to existing methods. I also like the AVA Python program package, which is well documented. Results demonstrate that AVA can capture important biological differences between vocalization e.g., between directed and undirected songs in birds, and identify similar syllables in mouse song. It can clusters syllables to types in adult zebra finches. The evidence for the lack of clusters in the mouse song are strong and convincing, and of important implications.

The principal weakness of the manuscript, in its present form, is that only insufficient evidence are provided to allow judging how AVA can perform in more difficult tasks, for which software like SAP is often used. For example, can AVA perform robust tutor-pupil similarity measurement in zebra finches? Can it identify clusters in young zebra finches? There is also no serious attempt to show replicability across different training sets. Once these concerns are address I feel that the manuscript should be appropriate for publication.

1. Detecting similarly as in Figure S5 across random repetition of the same zebra finch syllable spectrograms is not convincing enough. It is important to show how well can AVA performs when comparing tutor and pupil zebra finch songs. This testing should include examples of tutor pupil zebra finch songs (some with high similarity and others with lower similarity) should be plotted using UMAP projection as in Figure 6b.

2. It is relatively easy to detect clusters in adult zebra finch songs, but to study song learning it is often needed to cluster song syllables in young birds. SAP can often detect clusters in 50-day old birds. I wonder if AVA can detect clusters even earlier? This would be a very convincing demonstration to the power and usability of the new approach. Again, it is critical to show how AVA behaves with presented with more challenging tasks.

3. One issue that bothers me a lot is how the specific training of the algorithm might affect the outcomes. Say for example that lab 1 trained AVA with one dataset, and lab 2 trained AVA with a second dataset. But assume that both datasets were randomly sampled from the same population of birds. How comparable the results would be? For example will a similarity measurement of the same tutor and pupil would be comparable across the labs who trained AVA independently?

4. I like the "higher representational capacity" of the new method, but sometimes "with much wisdom comes much sorrow": higher representation capacity can potentially cause trouble if it makes the method too sensitive to things we do not care about. At this level, I would like to see some evidence for robustness to noise. For example, it should be easy to test how sensitive AVA is for small differences in recording conditions, say, to recording with in a sound attenuation chamber while door is open or closed?

*Reviewer #4:*

Low-dimensional learned feature spaces quantify individual and group differences in vocal repertoires.

The authors use variational autoencoders (VAEs) to learn a low-dimensional representations to spectrograms of bird song an mouse unltrasonic vocalizations (USVs). They find these representations of vocal behavior to be useful for studying social interactions and differences between strains of mice. Further investigations suggest that mouse USVs do not cluster as nicely as previously thought, and rather span a continuous manifold in feature space. Finally, VAEs trained on random snippets of the spectrogram highlight variability (and stereotypy) in zebra finch songs, in contrast to the unstructured (or highly variable) space of mouse USVs.

The proposed model operates in the frequency domain, consuming snippets of time-warped spectrograms. Given that one of the central claims of this paper is the superiority of unsupervised methods for feature extraction, I think these preprocessing steps warrant further consideration. For example, the 2D convolutions in the VAE must implicitly leverage the fact that neighboring frequencies are adjacent in the 128x128 ``images,' but does the choice of frequency spacing (mel-spaced for song birds vs linearly-spaced for mouse USVs) affect the learned representations? How important is the time-warping to downstream representations and analyses? The spectral preprocessing also complicates the ``shotgun' analysis in Figure 6. Each point in the VAE latent space corresponds to a window of time rather than a single frame. How do the projections change as you vary the window size?

Motivated in part by these concerns, some recent approaches like WaveNet (van den Oord et al., 2016) have directly modeled raw waveform data. The sampling rates used for human speech and music (16kHz) are lower than those necessary for USVs, but the same principles should apply. For example, a pre-trained WaveNet with minimal fine-tuning for song bird or mouse USV could yield a very competitive generative model of vocalizations and offer a different representation of this behavior. The comparison may be beyond the scope of this paper, but I think it is worthy of discussion.

Overall, this paper offers a nice application of nonlinear latent variable models to vocal behavior data. The techniques themselves are not particularly novel – variational autoencoders have been widely used in the machine learning community for over five years now – and the finding that learned features can outperform handcrafted ones has been shown across many domains. Given the wealth of works on sequential VAEs for time-series data, I think the novelty of the shotgun VAE is somewhat overstated. In my view, the main contribution lies in the codebase (I looked through the Github repo and was quite impressed!), the analysis pipeline, and the proof-of-concept. That is why I think it is especially important to assess the sensitivity of these results to various design choices that went into the pipeline, including the very first choices about how to preprocess the raw waveform data into time-warped and windowed spectrograms.

Other comments:

– Figure 1a: It's not clear why the length 32 vector is appearing as a square matrix here.

– Please label which dataset (song bird or mouse) the point clouds and spectrograms are coming from in all of the figures. A consistent color scheme could help.

– Figure 2f only has three solid lines. Where is the plot of explained variance in MUPET features by VAE features?

– The paragraph starting on line 97 paints an overly optimistic view of VAEs. Designing deep generative models that can reliably disentangle latent factors is still an active area of research, as is model selection.

– Figures 3, S4, and S5 suggest that nearest neighbor reconstruction with DeepSqueak (and other handcrafted features) is surprisingly bad. Are you using just the Euclidean distance in DeepSqueak feature space? Did you whiten the DeepSqueak features before computing distances? Can you explain why it DeepSqueak is failing so dramatically?

– Throughout, the spectrograms fail to indicate the time window or frequency bands.

– Figure 4a+b aim to show that SAP do not separate directed and undirected vocalizations as well as latent features do, but is this information simply not present in the first two PCs? A classification accuracy assessment would be more convincing.

– The 2D embedding in Figure 4f is confusing to me. Why not just show the full distance matrix from Figure S8, but with the lines to indicate which rows/columns belong to each mouse? That figure gives considerably more information than the tSNE embedding, in my opinion. In particular, it looks like there is a solid group of C57 mice that are very similar to DBA mice, as measured by MMD. The use of tSNE seems rather arbitrary and lossy. Moreover, the colors in Figure 4f convey little information beyond identity, when there seems to be lots of extra info about strain that could be conveyed.

– There are no axis labels or titles in Figure 5a-c, just random clouds of points.

[Editors’ note: further revisions were suggested prior to acceptance, as described below.]

Thank you for submitting your article "Low-dimensional learned feature spaces quantify individual and group differences in vocal repertoires" for consideration by *eLife*. Your article has been reviewed by 3 peer reviewers, including Jesse H Goldberg as the Reviewing Editor and Reviewer #1. and the evaluation has been overseen by Timothy Behrens as the Senior Editor. The following individuals involved in review of your submission have agreed to reveal their identity: Ofer Tchernichovski (Reviewer #2); Scott W Linderman (Reviewer #3).

The reviewers have discussed their reviews with one another, and the Reviewing Editor has drafted this to help you prepare a revised submission. This was deemed a strong resubmission and only one relatively minor issue related to Figure 7 is necessary for revision.

Essential revisions:

1. Figure 7 does not show clearly enough how the SG approach overcomes the problem of fused syllables (A and B). Figure 7c should somehow indicate the similarity in the fused AB vs A,B area. The issue is that it is not easy to see how the color code correspond to specific areas in the sonograms presented. An overlay panel could help here.

*Reviewer #1:*

In this paper, the authors addressed the reviewers' concerns and expanded extensively on the utility of variational autoencoder (VAE). The authors included an extra section discussing VAE 's capability in handling more complicated scenarios by studying the tutor and pupil song learning experiment. One can readily visualize the differences between tutor and pupil syllables via the latent embeddings. Although the latent features could be hard to interpret, one could view it as an initial exploratory analysis in identifying possible acoustic structure discrepancies. The authors also included additional data benchmarking latent features against conventional acoustics features for classification tasks and offered a more in-depth study comparing the clustering of song syllables using traditional acoustic features and VAE latent features. Moreover, they discussed the effect of time stretch and frequency spacing parameters on SAP features prediction and VAE's replicability issue for completeness.

The new Figure 7 showing tutor-pupil analyses is a welcome addition to the paper.

While it remains uncertain if this method will actually supersede others in quantifying finch and/or mouse datasets, this paper could, at minimum, provide a case study of advantages and disadvantages for using the VAE approach for vocalization datasets.

*Reviewer #2:*

This study applies an unsupervised learning approach for assessing acoustic similarity and for classifying animal vocalizations. Investigation focuses on mice vocalization and song learning in zebra finches. The method demonstrate an impressive capacity to map and compare vocal sounds in both species and to assess vocal learning. It has clear advantages upon existing methods. It is still an open question to what extent this approach can successfully capture vocal development during early stages of song learning. In particular, the learned latent features have no simple interpretation in production and perception of vocal sounds, which future studies will need to address.

Two remaining issues:

1. figure 7 does not show clearly enough how the SG approach overcomes the problem of fused syllables (A and B). Figure 7c should somehow indicate the similarity in the fused AB vs A,B area. The issue is that it is not easy to see how the color code correspond to specific areas in the sonograms presented. An overlay panel could help here.

2. The lack of song development analysis is still an issue.

*Reviewer #3:*

I thank the authors for their detailed responses. They have satisfactorily addressed all of my concerns.

---

## [Author Response]

[Editors’ note: the authors resubmitted a revised version of the paper for consideration. What follows is the authors’ response to the first round of review.]

The reviewers mostly agreed that the VAE is a potentially interesting approach to categorizing vocalization data, and there was enthusiasm about the codebase available in github. Some major problems that arose in review were (1) lack of strong behavioral insights; (2) lack of clarity about data pre-processing – and how this would affect results; and (3) concern about novelty given the widespread use of VAEs for similar problems.

We thank all four reviewers for their detailed and thoughtful critiques. In our substantially revised manuscript, we have attempted to address each of the concerns above by adding a major new analysis and numerous supplementary analyses that we believe help to round out our discussion and demonstrate the utility of the VAE as a tool for vocal analysis. In particular:

In response to a suggestion by Reviewer 3, we have included a significant new analysis comparing the similarity of zebra finch tutor and pupil song, now Figure 7. We believe these new results both demonstrate an improvement on existing methods and clearly point the way toward addressing point (1) above, since assessing similarity between tutor and pupil song constitutes a key method of behavioral analysis in birdsong research.

1. In response to suggestions by Reviewers 1, 3, and 4, we have performed several new analyses (included as Supplementary Figures 4, 5, 8, 11, 13, 14, 15, 17, and 18 and Supplementary Tables 1-3) that examine the effects of preprocessing choices, the reliability of training, and the quantitative performance of VAE features for downstream tasks. We discuss each of these more fully below; collectively, these work to address the concerns raised under point (2) above.

2. In response to concerns raised by Reviewers 2 and 4, we have revised the text to better place our work in the context of both vocal analysis and machine learning more broadly (point (3)). We clarify that, while the VAE has been applied to many kinds of problems, establishing its usefulness for a new audience is nonetheless valuable (as the reviewers’ comments attest) and necessitates a careful comparison with existing approaches to build confidence in the method.

Taken together, we believe these revisions both address each of the key concerns listed above and greatly improve the rigor and reach of the work. Replies to individual reviewers follow. Original comments are displayed in *italics*, our response, as here, in bold. Text added or altered in the revised manuscript is in blue.

We would, in principle, be open to considering a revised version of this manuscript if the relatively long list of concerns by reviewers 2 and 4 were adequately addressed and if the VAE approach could perform similarity-score metrics (as requested by reviewer 3).Reviewer #1:The authors proposed the use of variational autoencoder (VAE) to vocal recordings of model organisms (mouse and zebra finch) to better capture the acoustic features that are missed by conventional acoustic metrics. This manuscript explores the effectiveness of the VAE approach from two perspectives: (i) a motifs based clustering which seeks to match the acoustics data against several predetermined template and (ii) an unsupervised clustering based on randomly segmented vocal recordings. These approaches involve the generation of a collection of images from spectrograms that are then fed to variational encoders to estimate the number of latent variables. With these latent variables, the authors then employed UMAP to visualize the variation within the dataset. The analyses are well conducted and will be useful for broad range of scientists investigating animal vocalizations.i. From the zebra finch's discussion, this approach performs well in clustering the song syllables based on the four motifs predefined and at the same time could delineate the differences between directed and undirected songs as compared to previous acoustic metrics. While the authors provided a comparison between the ability of SAP acoustics features and VAE latent features in differentiating the directed and undirected birdsong, a comparison between the clustering of song syllables with the use of SAP features and VAE latent feature was not offered (Figure 5a). It would be interesting to see a side-by-side comparison of the 'Goodness of Clustering' metric in this VAE approach vs SAP.

We thank the reviewer for this suggestion. Results from this analysis are shown in Figure S13, which shows latent features produce better clusters than MUPET features, and Figure S14, which shows that SAP features also admit well-clustered clusters. In fact, on two of the three metrics we examine, SAP features produce better clusters than VAE features, while on the third, VAE features perform best. We believe these results illustrate two points:

1. When data cleanly separate into clusters — as they do in the bird data — multiple methods perform well, with the best-performing method depending on what measure one wants to optimize. That is, tight, well-isolated clusters satisfy multiple (semantically overlapping) notions of what clustering entails.

2. When data do not obviously cluster, the particular definition of clustering, as well as the assumptions of clustering methods, play a much larger role. As a result, MUPET features produce worse clusters than VAE features (by the metrics investigated). We unpack this more fully in our response to Reviewer 2 below.

ii. As for the randomly segmented vocal recordings, this method could generate a manifold where different acoustic features were situated in different regions and offer a continuous variability measure for all syllables. It is worth noting that the vocal recordings were randomly segmented based on the length of one syllable. What happens when the length is extended beyond that? Will the manifold produced look like the one shown in Figure 6?

This is an excellent question. Results are presented in Figure S17. Clearly, the manifolds are not identical, since (a) changing the size of the window introduces a change in the amount of data latent features must represent, and (b) the UMAP projection does not necessarily preserve distances or orientation, only local relationships. Nonetheless, features of the song like the linking note spiral and the song motif, including the correct ordering of syllables, are still present, consistent with the fact that the shotgun VAE does preserve local similarities in the data.

iii. As for the mouse vocal recordings, the VAE approach generates a continuum-like points cloud that performs reasonably well in differentiating different acoustic features of the mouse vocalizations albeit the lack of boundaries in separating them. Could the smooth variation of points be due to the sampling rate? The mouse vocalizations were sampled at a much higher rate (10x) as compared to bird vocalizations. I would expect a greater resolution for the mouse data and thus the VAE can capture more subtle differences between the vocalizations, yielding a continuum-like points cloud. Of course these sampling rate differences are justified because of the different spectral properties of birdsong and mouse USVs – but a simple sentence or two about how sampling rates may affect these analyses would be useful..

The answer to the specific question here is simple — while the raw data were sampled at different rates, the spectrogram data on which the VAE operates were of the same dimension (128 x 128 after preprocessing) in both bird and mouse — but raises an important issue regarding clustering: Is the quasi-continuum we see in Figure 5b,c the result of *some kind* of “fill in” behavior (as we ourselves raise in ll. 318-328)? Might there be some feature of the VAE that favors an unclustered latent representation?

This is an important reason for including both bird and mouse data, as we have done in the manuscript. Under the same set of analyses, bird data reliably and cleanly cluster, while the mouse data are, even under the most generous interpretation, ambiguous. Moreover, our data set sizes are not drastically larger than those used by other groups (again, we use MUPET’s own benchmark data), which argues against the idea that clustering is stymied by the size of the data set. But of course, as Reviewers 1and 2 both point out, UMAP is a visualization strategy, one not guaranteed to preserve metric information, which may lead to misleading displays. This is why we have performed all analysis on latent features (*not* UMAP representations) and attempted to tackle the problem from multiple directions, as we further explain in response to Reviewer 2 below.

Reviewer #2:Authors introduce a new method for behavioral analysis of vocalizations that supposedly improves on weaknesses of handpicked features, which is that they miss variability and introduce undesirable correlations. They apply VAEs to bird and mouse vocalizations and perform analysis of the latent space. They show that VAEs outperform handpicked features on some simple analysis tasks including time-resolved analysis of variability.Overall, even if it is correct that their method outperforms traditional-features based analysis, I don't think this insight is in any way relevant. Essentially, it is like saying: here is a bad way of doing things, and we offer one that is slightly better but still much worse than the gold standard. The problem is that there are many methods out there for doing things right from the start, so I cannot see a need for VAEs for the problems addressed. The problem with VAEs is that they are compressive (not representing all aspects of the data), which is why they are less desirable than methods that operate on equivalent representations of the raw data. If you are given money to buy groceries, why would you throw away part of it before you enter the store?

We believe it would be helpful to clarify here: our claim is not that traditional feature based analysis is a “bad” way of doing things. Clearly, this approach, whatever its limitations, has proven tremendously fruitful and continues to be the standard of practice in much of the vocalization community. Our goal is to suggest the VAE as an additional tool that offers improvements in several cases where feature-based methods have encountered difficulty, as our analyses show.

We must also confess some confusion as to which particular gold standard the reviewer is referencing, and this makes it difficult for us to reply on this point. Moreover, we are unsure how some degree of data compression is to be avoided. Feature-based methods are all a form of compression — they select some aggregate measures of the data while eschewing others — as are spectrograms, which throw away phase data. If nothing else, standard statistical tests each reduce the entire data to a single summary statistic for hypothesis testing.

To be clear: when a given analysis question can be answered by a method operating on raw data (with no intermediate compressed representation), we are not, *a priori*, advocating against this. But in the literature, there are many studies in which feature based methods remain the approach of choice (Derégnaucourt et al. (2005), Gaub et al. (2010), Hammerschmidt et al. (2012), Holy and Guo (2005), Chabout et al. (2015), Woehr (2014)), and we would argue that the present manuscript shows that VAE methods may offer advantages in these cases. It is difficult to say much more than this without a particular alternative analysis in view.

Overall it seems authors have not much to argue for other than their opposition to handpicked features. They pretend not much is out there other than these features for behavioral analysis, which does not reflect the wealth of methods out there. They have not even given appropriate credit to the real usefulness of the handpicked-feature approach, which is interpretability (which their method lacks). For example, one main utility of handpicked features is that some are learned. Songbirds have been shown to learn pitch from a tutor, so pitch is relevant. The same cannot be said about their latent features: which one of these do birds learn from a tutor?Also, correlations among features that authors criticize as a bug can be seen as a feature, it provides insights into the aspects of their vocalizations that animals cannot control independently.

We apologize for some lack of clarity on several points relevant to the reviewer’s concerns. In the interest of brevity and clarity, we have not attempted a comprehensive review of methods for the analysis of animal vocalization. Nor do we claim that the VAE could or should supersede all of these. Instead, we have focused on the feature-based approach, which overwhelmingly remains the standard of practice in the field and forms the basis of the analysis packages (SAP, MUPET, DeepSqueak) against which we provide extensive comparison.

We can also offer more clarity on two specific points the reviewer raises here:

1. While the learned latent features are not interpretable in the sense that they precisely correspond to constructs like pitch and maximum frequency well-known in signal processing, the point of panels 1a-c and Figure S1 is that this information is nonetheless present in the latent features in a straightforward way. That is, simple linear combinations of the latent features often correlate well (if imperfectly) with the traditional metrics of interest (Figure S1), in addition to capturing some information missed by these features.

The reviewer’s point about pitch is an important one, which we might rephrase as follows: it is not so much that juveniles learn pitch *per se* from a tutor as that juveniles are engaged in a complex, multi-feature problem of song learning, successful completion of which involves changes in pitch. That is, pitch forms only one axis along which behavior is changing, and our preference for describing the more complex learning problem in these terms is our own familiarity with this construct, both quantitatively and experientially. And while we do not wish to dispute the reasonability of this, as our new tutor/pupil song comparison shows (Figure 7), latent features precisely capture the similarity of the resulting songs, which has proven difficult to do with existing feature-based methods.

On decorrelation of features: As the reviewer notes, decorrelation of learned features is not, *a priori*, a bug, and the correlations among traditional features could possibly be put to interesting uses. In our analysis of feature decorrelation, our primary concern is for subsequent *statistical* testing, where nonlinear relationships among features violate the assumptions of many statistical tests and may lead to erroneous results. If traditional features were linearly related to one another, then a simple PCA or whitening operation, as is often done, would be sufficient to remedy this, but this does not appear to hold in real data (Figure S8). VAE features, which allow for nonlinear warping, avoid this. We have attempted to clarify this in the text (ll. 115-117):

“While correlations among features are not necessarily undesirable, they can complicate subsequent statistical testing because nonlinear relationships among features violate the assumptions of many statistical tests. VAE features, which allow for nonlinear warping, avoid this potential difficulty.”

I could also not detect any new behavioral insights. The only real strength in their manuscript I could detect is that their method allows them to visually segregate directed from undirected song (Figure 4b), vocalizations from different mice strains (Figure 4f), and songbird syllables (Figure 5a). Thus, their method could be a suitable pre-processing for clustering efforts. Also, on a positive note, their method also produces some interesting looking plots (Figure 6b) of to-be identified utility.

We apologize that what we believe to be novel behavioral insights were not better delineated. Briefly:

1. Mice produce stereotyped syllable repertoires that are stable across days (Figures 4f and S11, Table S3), but there is no evidence that the *sequences* in which these syllables are produced are stereotyped (Figure 6a).

2. We argue that the best account of USV distributions is an unclustered, quasicontinuum account (Figure 5; more on this below).

3. We show that zebra finch directed song is less variable in all syllable types, not just the tonal or harmonic syllables for which it has been established. That is, the VAE allows this to be observed and quantified for *all* syllables (Figures 4a-c), and this on sub-syllable time scales (6c-e).

4. We demonstrate a novel VAE-based method of computing tutor-pupil similarity in birdsong and quantify it in Figure 7.

L15 Authors claim that finding concise metrics remains a challenge, despite the large number of concise metrics ranging from Euclidean, Hamming, cosine, Wasserstein, Jensen-Shannon, Bhattacharia, Levenstein, ROUGE, etc. Surprisingly, their paper deals with the least concise metric imaginable, a deep autoencoder with thousands of parameters!

We apologize for the confusion. We are here using “metrics” in the more colloquial sense of “useful measurements,” not the mathematical construct the reviewer seems to be referencing. This has been fixed in the text, which now reads (ll. 16-17):

“[G]iven the variety and complex temporal structure of many behaviors, finding concise yet informative descriptions has remained a challenge.”

L19 They mention several powerful approaches to enable automatic analysis of vocalizations but cite none.

Thank you. This has been fixed (ll. 19-21).

L20 Given the wished-for generality of the problem they would like to study, it sounds strange that key to a successful approach would be a software package. Software packages are the last stage of analysis tool development.

The primary purpose of this work is to introduce and establish the VAE as a useful tool for vocal analysis. While we would contend that this is *conceptually* independent from any particular software implementation, we also note that, *practically*, code helps. We provide code as a means of disseminating these techniques, which Reviewers 3 and 4 applauded as providing additional value.

L26 [10] did not discover overnight consolidation of learned birdsong as claimed, but instead 'inappropriate consolidation', which is challenged by a more recent analysis in [26].

Thank you for this correction. The text has been edited to read (ll. 26-29):

“Collectively, these and similar software packages have helped facilitate numerous discoveries, including circadian patterns of song development in juvenile birds [10], …”

L29 Authors criticize 'correlations among features', claiming these could result in redundant characterizations of vocalizations. Implicitly they argue for features that do not correlate, e.g. independent component analysis. But again, no citations, no following up on the embarked ideas. Correlations could actually be a feature of the behavior itself (and quite interesting to study).

While we agree that correlations among features might provide interesting information, they can also reflect simple mathematical relationships among the feature sets chosen (x and x^3^ are strongly correlated by construction). However, as we have attempted to clarify under point 2 in our response above, such correlations can prove highly statistically problematic when attempting to test hypotheses about, e.g., changes in sets of vocal features across days, animals, or experimental conditions.

I do not believe that mouse USV syllables form a continuum of syllables. Because authors did not find clusters, this does not mean that they are not there. Rather than trying to find shortcomings (e.g. of their method or the number of samples analyzed), authors generalize from a negative finding to inexistence. By their rejection of vocal clustering, they also ignore previous results showing such clustering [18, 4, 48, 6, 16]. Quite audacious. Is the moon still there when I close my eyes?

We believe the reviewer is voicing here a warranted skepticism of our most controversial claim. And of course the absence of evidence is not evidence of absence. Yet, as we lay out in the manuscript (ll. 187-193), there is no absolute correct answer to the question of whether some particular data cluster, only more or less satisfying accounts. Reviewer 3 was convinced by our presentation, while others may remain skeptical. In favor of our conclusion, we note:

1. We argue from three directions: (a) the lack of apparent clustering for USVs as compared to bird song syllables (Figures 5a-b, S12); (b) the markedly worse performance of clustering algorithms (as quantified by several methods) in the mouse as compared to bird case (Figures 5c, S12-S15); (c) the existence of interpolating sequences within the data between qualitatively different USV syllables (Figures 5d, S16). Of these three, the last does not depend at all on VAE features and clearly fails for the bird song example.

2. Several of the works cited by ourselves and the reviewer ([4,6,16]) simply *assume* that mouse USVs cluster or that clustering offers a useful quantitative account of the behavior. One work [18] provides evidence of syllable clustering on the basis of a single-dimensional time/frequency curve description of USVs using a relatively small dataset of 750 syllables. We believe our analysis complements this approach by considering a more complete syllable description applied to much larger collections of syllables (17,400 syllables in Figures 5b-c). One of these works [48] reports a single bimodally-distributed acoustic feature, peak frequency, and refers to corresponding modes as clusters, but does not explicitly argue for clustering. To the best of our knowledge, our work provides the most comprehensive investigation of the USV syllable clustering hypothesis.

3. Many of the same works that lay claim to clustering use traditional features similar to MUPET features to cluster. To explicitly test whether MUPET features produce better behaved clusters than learned latent features, we added the analysis depicted in Figure S13. We found that MUPET features actually produce much worse clusters than latent features. In fact, when quantified by three unsupervised clustering metrics, MUPET feature-derived clusters are judged to be *less* clustered than moment-matched Gaussian noise.

4. On a more practical note, we have added Figure S15, which explores the consistency of syllable clusters under independent clusterings of different data splits. If cluster structure is well-determined by the data, we would expect syllables to be reliably segregated into the same clusters. This is indeed what we find for six clusters of zebra finch syllables, the number of clusters found by hand-labeling. However, for mouse syllables, we find poor consistency for more than two clusters.

5. Thus, while we cannot definitively prove the absence of clusters (nor do we know how to make such a claim rigorous), we have shown by a series of analyses that the USV data we analyze do not readily conform to a clustering account in the way bird song syllables do. Not only are the results markedly worse quantitatively, we show that there are sequences of USVs *in the data* that connect even the most disparate syllable types. We believe this calls into question the *utility* of a clustering account of USVs.

6. Finally, we have attempted in our original manuscript to nuance our conclusion (ll. 318-328), discussing multiple reasons we might have failed to find clusters. These include being open to the possibility of different results in other species/strains,

data set size, and a distinction between mathematical clustering and categorical perception.

In the caption of Figure 2d, authors state 'an ideal representation would exhibit minimal off-diagonal correlation', i.e., ICA is ideal. Why do we need VAEs then if ICA is ideal?

While ICA results in independence, it does so by choosing linear combinations of the original input vectors. The VAE can be viewed in one way as a nonlinear generalization of this demixing operation. So while ICA might be ideal if linear independence were the only desideratum, it must usually be paired with some form of dimension reduction, which the VAE will perform in a way that preserves more information.

Nonetheless, we agree that the point is confusing in the way it was stated. In the revised text, we have clarified (Figure 2 caption):

“When applied to the mouse USVs from a-c, the acoustic features compiled by the analysis program MUPET have high correlations, effectively reducing the number of independent measurements made.”

Caption Figure 2e, the representational capacity will depend on the number of features, which is not reported. Same for Figure 2f, the more features used, the more principal components will be needed, so this may be a trivial effect of unequal number of features.With regards to Figure 2f, it is not even clear from their analysis whether for a given fixed dimensionality, VAEs encode more variance than simple PCA, and if so, at what cost on memory (principal components vs auto-encoder network). For example, in the original Hinton paper in Science, the outcome of this analysis was rather surprising (VAEs are not clearly outperforming PCA in terms of representational capacity).

Thank you. We have added the number of features to the caption of Figure 2e. In Figure 2f, we agree, and have truncated the latent feature dimension to match the number of traditional features for a fairer comparison.

“Reducing the dimensionality of data with neural networks” (Hinton and Salakhutdinov, 2006) concerns deterministic autoencoders, which we do not believe are directly relevant to this work.

Last paragraph of Page 4, other than some pretty pictures (Figure S4) there is no (numerical) evidence for their claims of superiority of their latent features.

While we would argue that Figures 3 and S6 (previously S4) demonstrate clear qualitative improvements in nearest neighbor recovery, we have also included in our revised submission additional details regarding the failure of DeepSqueak features in identifying nearest neighbors (Figure S8) and several comparisons between the utility of VAE features and other feature sets for downstream classification tasks (Tables S1-S3). As those results demonstrate, VAE features clearly outperform handpicked feature sets.

L115-125 and Figure 4: This is an ill-advised analysis. Why would one choose SAP features to detect changes in song? It is obvious that change detection requires the most sensitive analysis possible, so why would one perform compression beforehand? Same goes for their latent features. Even if it is better than SAP, why would one choose it and not the raw data itself (and a suitable standard metric)? Same for L 126-L135 on data in mice.

We appreciate the reviewer raising this point. In situations with many more features than data points, learning in a generalizable way that does not severely overfit the data requires extra assumptions, for example sparsity or other informative prior beliefs. Dimension reduction can be seen as an attempt to cope with this fundamental limitation: if we can distill the relevant variation in our data into a much lower dimensional space, then we can re-enter the classical regime with more data points than features where off-the-shelf regression and classification are known to work well.

We also note that many highly influential analyses of song learning have relied on SAP features to detect changes in song (Tchernichovski et al. “Dynamics of the vocal imitation process: how a zebra finch learns its song” (2001); Ravbar et al. “Vocal exploration is locally regulated during song learning” (2012); Vallentin et al. "Inhibition protects acquired song segments during vocal learning in zebra finches." (2016); and many others).

L 173-L187: Authors argue about clustering failures of MUPET using UMAP representations, ignoring the fact that UMAP provides a faulty account of true distance. Their analysis of latent features is a step in the right direction, but falls short of analysis of the raw data (or an equivalent representation, e.g. https://asa.scitation.org/doi/abs/10.1121/1.4731466 and others).

First, all of our clustering comparisons are performed on features (MUPET, SAP, VAE, etc.), with UMAP only used for visualization, as stated in the manuscript. And we agree with the reviewer that the UMAP representation cannot tell the whole story. Our UMAP plot simply serves the purpose of providing a contrast between mouse and bird syllables and motivating further analyses.

As for the reviewer’s other concern, we *think* the reviewer is suggesting that clustering might better be performed on the raw data, and that failure to find clusters in the USV case might be attributable to dimension reduction. To this objection, we first note that the studies we cite that report clustering of USVs (with which the reviewer appears to agree) all use a feature-based clustering approach. It is hard to see how both (a) featurebased clustering methods are illegitimate for USVs and (b) reports of USV clustering based on features can be trusted.

Moreover, we are unsure what clustering based on raw data in this case might mean. Even our spectrograms, which ignore phase data, comprise tens of thousands of data points per observation, and clustering data in dimensions this high is known to be borderline meaningless without strong priors or dimension reduction (cf. Aggarwal, Hinneberg, Keim (2001); Bouveyron and Brunet-Saumard (2014)). The reviewer may have one of these specific techniques in mind, but we also note that the highly structured nature of spectrograms implies that invariances in the interpretation of the data (like small pixel shifts in time) may not be respected by a naive clustering of raw data.

L214: Their method is not useful for analyzing moment-by-moment variability of song, because they need to pre-process songs by 'warping each in time to account for well-documented differences in rhythm and tempo', which is the only problem that would complicate a more rigorous analysis.

We apologize for some lack of clarity in our description. While time warping does improve our ability to align and thus compare syllables, the difference is not nearly so large as might be imagined. Figure S18 reproduces Figures 6d-e without time warping applied. More to the point, we are a bit confused by the contention that, apart from time warping, the moment-by-moment analysis of variability in song is trivial. We were unaware of any published results on this problem. Indeed several other reviewers noted this new analysis as a real advance beyond the current state of the art.

Reviewer #3:This manuscript presents a new implementation of variational autoencoder machine learning algorithm (VAE) for the analysis of animal vocalization. The new method is impressive and powerful compared to existing methods. I also like the AVA Python program package, which is well documented. Results demonstrate that AVA can capture important biological differences between vocalization e.g., between directed and undirected songs in birds, and identify similar syllables in mouse song. It can clusters syllables to types in adult zebra finches. The evidence for the lack of clusters in the mouse song are strong and convincing, and of important implications.The principal weakness of the manuscript, in its present form, is that only insufficient evidence are provided to allow judging how AVA can perform in more difficult tasks, for which software like SAP is often used. For example, can AVA perform robust tutor-pupil similarity measurement in zebra finches? Can it identify clusters in young zebra finches? There is also no serious attempt to show replicability across different training sets. Once these concerns are address I feel that the manuscript should be appropriate for publication.

We appreciate the reviewer’s positive assessment of the work. Based on the reviewer’s suggestions, we have implemented several new analyses detailed below. We hope the reviewer agrees that these significantly enhance the work by demonstrating the robustness and applicability of the VAE.

1. Detecting similarly as in Figure S5 across random repetition of the same zebra finch syllable spectrograms is not convincing enough. It is important to show how well can AVA performs when comparing tutor and pupil zebra finch songs. This testing should include examples of tutor pupil zebra finch songs (some with high similarity and others with lower similarity) should be plotted using UMAP projection as in Figure 6b.

This is an excellent suggestion. To address this, we have added an entirely new analysis (Figure 7) to test how well the VAE-based approach performs detecting tutor and pupil song similarity. In this analysis we take 10 pairs of zebra finch tutors and pupils and inspect the VAE’s learned representations using both syllable-based and shotgun VAE approaches. For both approaches, we find highly similar learned representations for the tutor and pupil pairs (Figure 7a,d). Additionally, we find that the VAE represents finer scale variations in the quality of song copying (Figure 7b,e) and demonstrate that the maximum mean discrepancy (MMD) metric proposed earlier in the manuscript is effective for quantifying the quality of song copying. We believe this constitutes a novel advance on current practice for quantifying tutor/pupil similarity in zebra finch song.

2. It is relatively easy to detect clusters in adult zebra finch songs, but to study song learning it is often needed to cluster song syllables in young birds. SAP can often detect clusters in 50-day old birds. I wonder if AVA can detect clusters even earlier? This would be a very convincing demonstration to the power and usability of the new approach. Again, it is critical to show how AVA behaves with presented with more challenging tasks.

We agree with the reviewer that the ability to cluster adult songs is no surprise. We also agree that identifying syllables in young birds is a much more interesting test. Indeed, as part of a separate collaboration between our labs, we have been pursuing these questions. In practice, we find the major obstacle to detecting syllable clusters in very young birds is our ability to reliably segment syllables. In order to bypass this difficulty, we are currently using the shotgun VAE approach for analyzing the song of very young birds. Encouragingly, despite the segmenting difficulties, we find that the syllable-based VAE analysis identifies some structure in the song of a 50-day-old juvenile, including a clear separation between calls and subsong.

Largely the same structure is evident five days earlier, when the bird is 45 days old.

3. One issue that bothers me a lot is how the specific training of the algorithm might affect the outcomes. Say for example that lab 1 trained AVA with one dataset, and lab 2 trained AVA with a second dataset. But assume that both datasets were randomly sampled from the same population of birds. How comparable the results would be? For example will a similarity measurement of the same tutor and pupil would be comparable across the labs who trained AVA independently?

We share this concern and note two enframing points: First, replicability will be a function of what question is subsequently asked of the VAE. For instance, any rotation of features in latent space is equally valid, so there is no guarantee – in fact, it is unlikely – that “latent feature 1” will be the same across training runs. However, other analyses that are invariant to this degree of freedom may replicate across retraining. Second, producing replicable results across training runs of neural network models constitutes an interesting and important research question in machine learning, one on which we have a separate manuscript in preparation. That is, while we believe this is an important and interesting point, a proper treatment of it is beyond the scope of this work.

To address the reviewer’s suggestion, we added Figure S5 to investigate the similarity of latent features across training runs, both using identical training data and disjoint subsets of data. For mouse USV syllables, we find very good correspondence in both cases in terms of pairwise distances and linear predictions of one set of latent features from the other. For zebra finch syllables, we find poorer pairwise distances but fairly accurate linear predictions in both cases. Taken together with Figure S15, which shows near-perfect identification of zebra finch syllable clusters, we believe the poorer performance of zebra finch syllable alignment compared to mouse syllable alignment is driven by the inconsistency of the between-cluster structure, the relative positions and orientations of syllable clusters, which is underdetermined by the data.

4. I like the "higher representational capacity" of the new method, but sometimes "with much wisdom comes much sorrow": higher representation capacity can potentially cause trouble if it makes the method too sensitive to things we do not care about. At this level, I would like to see some evidence for robustness to noise. For example, it should be easy to test how sensitive AVA is for small differences in recording conditions, say, to recording with in a sound attenuation chamber while door is open or closed?

This is well put and an important point to address. Because the VAE is trained to capture the distribution in the training data, it will represent the largest sources of variation in the features it learns. Unfortunately, this method has no built-in notion of “relevant” versus “irrelevant” features. However, in the experiments reported here, systematically varying each of the latent features does produce changes in syllable shapes that appear meaningful. That is, they do not appear to capture artifacts. However, if artifacts were present non-uniformly in the data, it is very likely that the VAE would encode this. This could be addressed in two ways: (a) better data preprocessing, which would remove the artifact; (b) approaches like domain adaptation and other methods of de-biasing machine learning that work by identifying sources of variance (like race or gender in lending decisions) that systems should ignore (see, for example, Louizos et al. “The variational fair autoencoder” (2015)). We believe this will be an interesting avenue to pursue in future work.

Reviewer #4:Low-dimensional learned feature spaces quantify individual and group differences in vocal repertoires.The authors use variational autoencoders (VAEs) to learn a low-dimensional representations to spectrograms of bird song an mouse unltrasonic vocalizations (USVs). They find these representations of vocal behavior to be useful for studying social interactions and differences between strains of mice. Further investigations suggest that mouse USVs do not cluster as nicely as previously thought, and rather span a continuous manifold in feature space. Finally, VAEs trained on random snippets of the spectrogram highlight variability (and stereotypy) in zebra finch songs, in contrast to the unstructured (or highly variable) space of mouse USVs.The proposed model operates in the frequency domain, consuming snippets of time-warped spectrograms. Given that one of the central claims of this paper is the superiority of unsupervised methods for feature extraction, I think these preprocessing steps warrant further consideration. For example, the 2D convolutions in the VAE must implicitly leverage the fact that neighboring frequencies are adjacent in the 128x128 ``images,' but does the choice of frequency spacing (mel-spaced for song birds vs linearly-spaced for mouse USVs) affect the learned representations? How important is the time-warping to downstream representations and analyses? The spectral preprocessing also complicates the ``shotgun' analysis in Figure 6. Each point in the VAE latent space corresponds to a window of time rather than a single frame. How do the projections change as you vary the window size?

This is an excellent point. In practice, preprocessing can be as important as the model itself in deriving usable results. In response to the reviewer’s questions, we have performed several new analyses, including:

1. In Figure S4, we compare latent representations of mel vs. linear frequency spacing for zebra finch syllables and stretched vs. non-stretched syllables for both mouse and zebra finch. For zebra finch syllables, we find both comparisons produce latent features that encode hand-picked features to a similar extent and also display reasonable consistent pairwise distances (R^2^≈0.4). For mouse syllables, we find that the time stretch does provide a small but consistent benefit in terms of latent features representing hand-picked features and also fairly consistent pairwise distances with or without the time stetch (R^2^=0.69).

2. We have repeated the shotgun VAE analysis for temporal windows of varying size. The results are in S17. While there are obvious but trivial differences (overall rotations), major features like the linking note and song motif of bird song maps are reproduced, as well as the coarse structure of mouse USVs. However, fine details, particularly in the non-metric UMAP projection, do change, and these highlight the need, as mentioned in our response to Reviewer 3 above, for identifying downstream analyses that do not depend on small details of these maps.

Motivated in part by these concerns, some recent approaches like WaveNet (van den Oord et al., 2016) have directly modeled raw waveform data. The sampling rates used for human speech and music (16kHz) are lower than those necessary for USVs, but the same principles should apply. For example, a pre-trained WaveNet with minimal fine-tuning for song bird or mouse USV could yield a very competitive generative model of vocalizations and offer a different representation of this behavior. The comparison may be beyond the scope of this paper, but I think it is worthy of discussion.

We agree that this is both a really interesting direction and worth discussing. We have added the following text to the manuscript (ll. 341-342):

“We also note that great progress in generating raw audio has been made in the past few years, potentially enabling similar approaches that bypass an intermediate spectrogram representation (va den Oord et al., 2016, Kong et al., 2020).”

Overall, this paper offers a nice application of nonlinear latent variable models to vocal behavior data. The techniques themselves are not particularly novel – variational autoencoders have been widely used in the machine learning community for over five years now – and the finding that learned features can outperform handcrafted ones has been shown across many domains.

We agree. Our purpose here is to introduce these methods to many in the neuroscience community who’ve not been aware of their utility and to demonstrate that they produce useful and novel results in this particular domain.

Given the wealth of works on sequential VAEs for time-series data, I think the novelty of the shotgun VAE is somewhat overstated.

This is fair. Incorporating some of the sequential VAE methods is an interesting direction for future work. We have modified the text to temper the claim of novelty for this approach. Nevertheless, we feel the subsampling-based approach offers an intuitive and accessible single-parameter stand-in for what are often substantially more powerful (but complicated) time series models.

In my view, the main contribution lies in the codebase (I looked through the Github repo and was quite impressed!), the analysis pipeline, and the proof-of-concept. That is why I think it is especially important to assess the sensitivity of these results to various design choices that went into the pipeline, including the very first choices about how to preprocess the raw waveform data into time-warped and windowed spectrograms.

We concur. To recap our efforts along these lines, we have included S4, which investigates the effect of spectrogram frequency spacing and time stretching on the resulting latent features.

Other comments:– Figure 1a: It's not clear why the length 32 vector is appearing as a square matrix here.

Thank you. This has now been fixed.

– Please label which dataset (song bird or mouse) the point clouds and spectrograms are coming from in all of the figures. A consistent color scheme could help.

Thank you for this suggestion. Green and brown/orange are used to indicate mouse vocalization (as in Figure 4d) and purple and cyan are used to indicate zebra finch vocalization (as in Figure 4a).

– Figure 2f only has three solid lines. Where is the plot of explained variance in MUPET features by VAE features?

We regret the confusion. In the original submission, only two lines were shown because there was a single set of latent features corresponding to the same syllables as both the MUPET and DeepSqueak features. In response to comments from reviewer 2, however, we have now truncated the single set of latent features to match the dimensions of the MUPET and DeepSqueak feature sets, resulting in separate lines. The caption for the figure now contains additional text:

“Latent features with colors labeled “MUPET” and “DeepSqueak” refer to the *same* set of latent features, truncated at different dimensions corresponding to the number of acoustic features measured by MUPET and DeepSqueak, respectively.”

– The paragraph starting on line 97 paints an overly optimistic view of VAEs. Designing deep generative models that can reliably disentangle latent factors is still an active area of research, as is model selection.

The reviewer is correct that many of these problems still constitute pressing research questions. We have omitted this mention of disentangling, instead adding a reference to (Dai et al., 2018), which specifically analyzes sparse latent representations in VAEs.

- Figures 3, S4, and S5 suggest that nearest neighbor reconstruction with DeepSqueak (and other handcrafted features) is surprisingly bad. Are you using just the Euclidean distance in DeepSqueak feature space? Did you whiten the DeepSqueak features before computing distances? Can you explain why it DeepSqueak is failing so dramatically?

This is an excellent question. For Figures 3 and S7 (previously S5), we used Euclidean distance with standardized but not whitened DeepSqueak features (ll. 423-430). We have added Figure S8 to investigate the failure of the DeepSqueak feature set, in particular, in returning reasonable nearest neighbors by constructing alternative versions of the feature set. In one version we whitened DeepSqueak features, in the next we considered the feature set without the three features most poorly predicted by latent features (from Figure S1), and lastly, we considered the linear projection of DeepSqueak feature space most predictive of the VAE’s features. All three feature sets return more suitable nearest neighbors for *some* of the query spectrograms shown from Figure 3, but the improvements are slight and none are as close as the latent feature nearest neighbors. In particular, the poor performance of the last feature set, an attempt to bring DeepSqueak features in line with latent features, suggests that DeepSqueak features are not sufficient to capture the full range of mouse USV syllables.

– Throughout, the spectrograms fail to indicate the time window or frequency bands.

While we agree that this information is useful, we are somewhat concerned about the additional visual clutter to figures that need to display many spectrograms. These numbers are clearly stated in the methods.

– Figure 4a+b aim to show that SAP do not separate directed and undirected vocalizations as well as latent features do, but is this information simply not present in the first two PCs? A classification accuracy assessment would be more convincing.

We thank the reviewer for this suggestion. To address this, we performed additional analyses using different feature sets for downstream classification tasks (Tables S1-3). In particular, classifiers trained to predict social context performed better when using latent features than when using SAP features (Tables S1).

– The 2D embedding in Figure 4f is confusing to me. Why not just show the full distance matrix from Figure S8, but with the lines to indicate which rows/columns belong to each mouse? That figure gives considerably more information than the tSNE embedding, in my opinion. In particular, it looks like there is a solid group of C57 mice that are very similar to DBA mice, as measured by MMD. The use of tSNE seems rather arbitrary and lossy. Moreover, the colors in Figure 4f convey little information beyond identity, when there seems to be lots of extra info about strain that could be conveyed.

We appreciate the suggestion. We have elected to retain the t-SNE embedding in the main figure, since it makes certain features more available at a glance, and provides an analogous presentation to our visualizations of syllables, but we have also added Figure S11 to include the full MMD matrix with easily visualized individual and strain information.

– There are no axis labels or titles in Figure 5a-c, just random clouds of points.

Thank you. We have fixed this.

[Editors’ note: what follows is the authors’ response to the second round of review.]

Essential revisions:1. Figure 7 does not show clearly enough how the SG approach overcomes the problem of fused syllables (A and B). Figure 7c should somehow indicate the similarity in the fused AB vs A,B area. The issue is that it is not easy to see how the color code correspond to specific areas in the sonograms presented. An overlay panel could help here.

We appreciate the suggestion. Figure 7e now contains an inset illustrating similarity between the fused AB syllable from the pupil and the individual A and B syllables from the tutor. When assessed by MMD, the VAE trained on syllables judges the fused syllable to be distinct from both A and B, while the shotgun VAE shows a clear similarity (dark line) between A, B, and AB.